# Near-Optimal Collaborative Learning in Bandits

**Clémence Réda**
Université Paris Cité,
Inserm, NeuroDiderot,
F-75019 Paris, France
`clemence.reda@inria.fr`

**Sattar Vakili**
MediaTek Research,
Cambourne Business Park,
CB23 6DW, United Kingdom
`sattar.vakili@mtkresearch.com`

**Emilie Kaufmann**
Université de Lille,
CNRS, Inria, Centrale Lille,
UMR 9189 CRIStAL,
F-59000 Lille, France
`emilie.kaufmann@univ-lille.fr`

## Abstract

This paper introduces a general multi-agent bandit model in which each agent is facing a finite set of arms and may communicate with other agents through a central controller in order to identify –in pure exploration– or play –in regret minimization– its optimal arm. The twist is that the optimal arm for each agent is the arm with largest expected *mixed* reward, where the mixed reward of an arm is a weighted sum of the rewards of this arm for *all agents*. This makes communication between agents often necessary. This general setting allows to recover and extend several recent models for collaborative bandit learning, including the recently proposed federated learning with personalization [30]. In this paper, we provide new lower bounds on the sample complexity of pure exploration and on the regret. We then propose a near-optimal algorithm for pure exploration. This algorithm is based on phased elimination with two novel ingredients: a data-dependent sampling scheme within each phase, aimed at matching a relaxation of the lower bound.

## 1 Introduction

Collaborative learning is a general machine learning paradigm in which a group of agents collectively train a learning algorithm. Some recent works have investigated how agents can efficiently perform sequential decision making in a collaborative context [36]. In particular, Shi et al. propose an interesting setting to tackle collaborative bandit learning when some level of personalization is required [30]. Personalization leads to the twist that each agent should play the best arm in a *mixed model* which is obtained as a combination of her *local model* with the local model of other agents. In this work, we introduce a more general model retaining this idea, that we call the weighted collaborative bandit model.

In this model, there are $M$ agents and a finite number of $K$ arms. When agent $m$ samples arm $k$ at time $t$, she gets to observe a *local reward* $X_{k,m}(t)$, which is drawn from a 1-sub-Gaussian [1] distribution of mean $\mu_{k,m}$, independently from past observations and from other agents' observations. However, this agent does not necessarily seek to maximize her local reward, but rather some notion of *mixed reward*, related to the utility of that arm for other agents. More specifically, we assume that agents share a *known* weight matrix $W := (w_{n,m})_{n,m} \in [0,1]^{M \times M}$, such that $\sum_{n \in [M]} w_{n,m} = 1$

---

[1]A random variable $X$ is said to be $\sigma^2$-sub-Gaussian if, for any $\lambda \in \mathbb{R}$, $\ln(\mathbb{E}[e^{\lambda(X - \mathbb{E}[X])}]) \le \sigma^2 \lambda^2/2$.

for all $m \in [M]$, where $[n] := \{1, 2, \ldots, n\}$. The mixed reward at time $t$ for agent $m$ and arm $k$ is defined as a weighted average of the local rewards across all agents $X'_{k,m}(t) := \sum_{n=1}^{M} w_{n,m} X_{k,n}(t)$, and its expectation, called the expected mixed reward, is

$$\mu'_{k,m} := \sum_{n=1}^{M} w_{n,m} \mu_{k,n} .$$

We denote by $k_m^\star := \arg\max_{k \in [K]} \mu'_{k,m}$ the arm with largest expected mixed reward for agent $m$, assumed unique. Besides the degenerated case in which $w_{n,m} = \mathbb{1}(n = m)$, in which each agent is solving their own bandit problem in isolation, the agents need to *communicate* ; *i.e.*, to share information about their local rewards to other agents for everyone to be able to estimate their expected mixed rewards. A strategy for an agent is defined as follows: at each time $t$, each agent $m$ samples an arm $\pi_m(t)$, based on the available information, and then observes a noisy local reward from this arm. She has also the option to communicate information (*e.g.*, empirical means of past local observations) to a central controller (or server), which will broadcast this information to all other agents. Just like in any multi-armed bandit model, several objectives may be considered, either related to maximizing (mixed) rewards –or, equivalently, minimizing regret– or identifying the best arms, while maintaining a reasonably small communication cost.

The weighted collaborative bandit model encompasses different frameworks previously studied in the literature. Notably, the paper [30] studies a special case in which, given a level of personalization $\alpha \in [0,1]$, the mixed reward is an interpolation between $\mu_{k,m}$ and the average of local rewards $\frac{1}{M} \sum_{n=1}^{M} \mu_{k,n}$ which amounts to choosing $w_{m,m} = \alpha + \frac{1-\alpha}{M}$ and $w_{n,m} = \frac{1-\alpha}{M}$ for $n \neq m$. The authors consider the objective of minimizing the regret while minimizing for the number of communication rounds, under the name federated multi-armed bandit with personalization[2]. In this paper, we mainly focus on the counterpart pure exploration problem in which agents should collaboratively identify their *own* optimal arm in terms of expected mixed reward, with high confidence, and using as few exploration rounds as possible. This extends the well-studied fixed-confidence best arm identification problem [12, 14, 20] to the weighted collaborative bandit setting.

Another related setting is collaborative pure exploration [18, 32, 6, 35], which considers $M$ agents solving the *same* best arm identification problem. Most of these papers propose algorithms to solve this problem, while [32] and [21] also prove lower bounds on sample complexity for the fixed-confidence and the fixed-budget best arm identification. Unlike our framework, [6] consider asynchronous agents, which can only sample at some times, whereas [35, 36] consider agents which can only communicate to some of the other agents. The goal of collaborative pure exploration is to reduce sample complexity at the cost of some communication rounds. Our model recovers the synchronous setting when considering $\mu_{k,m} = \mu_{k,m'}$ for all arm $k$ and agents $m, m'$ and $W = Id_M$.

Besides collaborative learning, the work of [34] considers a similar weighted model but in a different, contextual bandit setting in which a central controller chooses in each round an arm for the unique agent (corresponding to a sub-population) that arrives, and aim at minimizing regret. Finally, the paper [29] considers a pure exploration task in which the value of an arm is the weighted average of its utility for $M$ distinct populations (agents). In this setting $w_{n,m} = \alpha_n$, so that all agents have a common best arm, but the proposed algorithms do not aim at a low communication cost.

Collaborative learning in our general weighted model is also interesting beyond these examples. In particular, the work of [30] mentioned possible applications to recommendation systems, for which one may want to go beyond uniformly personalized learning. The "personalization" part means that we favour the local rewards of agent $m$ over other agents' observations in the identification of optimal arm $k_m^\star$. But the introduction of a general weight matrix $W$ in our framework allows *any* agent $m$ to consider *any* linear combination of the other agents' observations, and then, different degrees of personalization across agents.

Such a setting could also be appropriate for adaptive clinical trials on $K$ therapies, run by $M$ teams who have access to different sub-populations of patients. In this context, each sub-population is typically aiming at finding their (local) best treatment. However, solving their best arm identification in isolation may have a large sample complexity. If one is willing to assume that we have a

---

[2]Their setting, as well as ours, is neglecting some challenges typically addressed in (centralized) federated learning, such as privacy issues or dealing with communication interruption, this is why we prefer naming our framework "collaborative learning".

weight matrix $W$ for which the best mixed arm of each agent coincides with its local best arm, *i.e.* $k_m^\star = \arg\max_{k \in [K]} \mu_{k,m}$ for any agent $m$, then solving best arm identification in the weighted collaborative bandit could have a much smaller sample complexity due to the sharing of information, while allowing the different clinical centers to communicate only once in a while. A first possibility to build such a weight matrix is to rely on a clustering of the different sub-populations so that the $\mu_{k,m}$ is supposed to be close to $\mu_{k,m'}$ (but not necessarily equal) for all $k$ and all agents $m, m'$ in the same cluster. Denoting by $C_m$ the cluster to which agent $m$ belongs, by setting $w_{m,n} = \mathbb{1}(C_m = C_n)/|C_m|$ we would have the mixed mean of each agent be very close to their local means. Another possibility is to rely on a similarity function $S$ between sub-populations and define for all $n, m$ $w_{m,n}$ to be proportional to $S(m,n)$. This similarity function could be obtained prior to the learning phase by computing the similarities between subpopulation biomarkers, for instance.

**Related work** Collaborative bandit learning has recently sparked wide interest in the multi-armed bandit literature. While some works do not deal with personalization [11, 31, 26], others have for instance studied the integration of arm features in a modified setting with personalization [19]. An interesting kernelized collaborative pure exploration problem was recently studied by [10]. In their model, both agents and arms are described by feature vectors, and there is a known kernel encoding the similarity between the mean reward of each (agent,arm) pair. This independent work follows a similar approach as ours and also propose a near-optimal phased elimination algorithm inspired by a lower bound, but the models and related lower bounds are significantly different.

In bandits working in collaboration, the need for a small communication cost indeed makes algorithms based on *phased eliminations* appealing. In such algorithms, agent(s) maintain a set of a *active* arms that are candidate for being optimal, and potentially eliminate arms from this set at the end of each sampling phase. Adaptivity to the observed rewards (and, in our case, communication) is only needed between sampling phases, which are typically long. This type of structure has been used in various bandit settings, both for regret minimization or pure exploration objectives [2, 18, 5, 13, 30, 3]. In some of these algorithms, including the one in [30] which motivates this paper, the number of samples gathered from an arm which is active in some phase $r$ is fixed in advance. We believe that going beyond such a deterministic sampling scheme is crucial to achieve optimal performance with phased algorithms. In order to achieve (near-)optimality, other phased algorithms rely on computing an oracle allocation from the optimization problem associated with a lower bound on the sample complexity [13, 10]. In these works, based on this allocation, a total number of samples to collect in the current phase is computed, which depends on the identity of the surviving arms. The distribution of samples across arms for the current round is proportional to the oracle allocation, and is obtained through a rounding procedure. Compared to these works, our algorithm will show three distinctive features: an allocation inspired by a *relaxation* of the lower bound, which does not only depend on the identity of surviving arms, and an alternative to the rounding procedure.

**Contributions** The authors of [30] exhibit an algorithm using phased elimination for federated bandit learning with personalization, and prove a logarithmic regret bound, whose dependency in the parameters of the problem is conjectured to be sub-optimal. They also propose a heuristic improvement, based on a more adaptive exploration within each phase. In this work, we take a step further in identifying the problem-dependent complexity of bandit learning in our novel weighted collaborative bandit model –which includes the setting of [30] as a special case– both from the pure exploration and the regret perspective. We propose new information theoretic lower bounds, and a recipe to design algorithms (nearly) matching those for pure exploration. Our main algorithmic contribution is for weighted collaborative best arm identification. Our phased elimination-based algorithm achieves minimal exploration cost up to some logarithmic multiplicative factors, while using a constant amount of communication rounds. It relies on a novel data-dependent sampling scheme, which renders its analysis trickier. The structure of the algorithm can easily be extended to Top-$N$ identification, that is, where each agent has to identify her own $N$ best arms (instead of her best for $N = 1$) with respect to mixed rewards. We further compare our novel regret lower bound to what was conjectured in [30] for the particular case of federated learning with personalization.

**Notation** For both best arm identification and regret minimization, our complexity terms feature the (mixed) gaps of each agent $m$ and arm $k$, defined by

$$\Delta'_{k,m} := \begin{cases} \mu'_{k_m^\star,m} - \mu'_{k,m} & \text{if } k \neq k_m^\star \,, \\ \min_{k \neq k_m^\star} \Delta'_{k,m} & \text{otherwise} \,. \end{cases}$$

## 2   Collaborative Best-Arm Identification

The goal of collaborative best-arm identification is that each agent $m$ identifies its optimal arm $k_m^\star$ by sampling the arms as little as possible and with few communication rounds. Formally, a collaborative Best Arm Identification (BAI) algorithm consists of a sampling rule $\pi_m$ for each agent $m$, such that, at time $t$, either arm $\pi_m(t) \in [K]$ is sampled by $m$, or $\pi_m(t) = 0$ ; in that case, instead of picking an arm at time $t$, we allow the agent to remain idle, and not to select an arm. [3] Similarly to the communication model studied in [32, 30], communication only happens at the end of local sampling rounds for all agents, when all agents are idle at the same time. Besides the sampling rule, the BAI algorithm uses a stopping rule $\tau$ which determines when exploration is over *for all agents*. The end of exploration is decided by the central server. Then, at time $\tau$, each player outputs a guess for its optimal arm with respect to mixed rewards, denoted by $\hat{k}_m$.

Our goal is to construct a $\delta$-correct strategy $\mathcal{A} = (\pi, \tau, \hat{k})$, which satisfies, for any model $\mu \in \mathbb{R}^{K \times M}$,

$$\mathbb{P}_\mu^\mathcal{A} \left( \forall m \in [M], \hat{k}_m = k_m^\star \right) \geq 1 - \delta ,$$

while achieving a small *exploration cost* (e.g. in high probability or in expectation)

$$\mathrm{Exp}_\mu(\mathcal{A}) := \sum_{m=1}^{M} \sum_{k=1}^{K} N_{k,m}(\tau) ,$$

where $N_{k,m}(t) := \sum_{s=1}^{t} \mathbb{1}_{(\pi_m(s)=k)}$ is the number of selections of arm $k$ by agent $m$ up to time $t$, and a small *communication cost*, defined as

$$\mathrm{Com}_\mu(\mathcal{A}) := \sum_{t=1}^{\tau} \mathbb{1}(\mathcal{I}_t) ,$$

where $\mathcal{I}_t$ is the event that some information is shared between agents at round $t$.

In our setting, we do not put constraints on the type of information that is exchanged in each communication round –which can be interesting when we consider privacy issues [11, 36]– nor on the lengths of the messages. Each communication round has a unit cost. In a communication round, all agents send messages to the central server (*e.g.*, estimates of their local means) and the server can send back arbitrary quantities or instructions (*e.g.*, how many times each arm should be sampled in the next exploration phase, and when to communicate next).

Moreover, contrary to the works of [18, 32] on collaborative learning, we do not look at strategies explicitly minimizing for the number of communication rounds. Instead, our approach consists in proving a lower bound on the smallest possible exploration cost of a $\delta$-correct algorithm which would communicate at every round ; and then, finding an algorithm for which exploration cost matches this lower bound, while suffering a reasonable communication cost.

## 3   Lower Bound

We prove the following lower bound on the exploration cost of an algorithm in which all agents communicate to the central server their latest observation as soon as they received it. It holds for Gaussian rewards with variance $\sigma^2 = 1$ , meaning that the reward from an arm $k$ observed at time $t$ by agent $m$ will be $X_{k,m}(t) = \mu_k + \varepsilon_t$ , where $\varepsilon_t \sim \mathcal{N}(0,1)$ . We further assume that the weight matrix $W$ satisfies $w_{m,m} \neq 0$ for any agent $m \in [M]$ .

**Theorem 1.** *Let $\mu$ be a fixed matrix of means in $\mathbb{R}^{K \times M}$. For any $\delta \in (0, 1/2]$, let $\mathcal{A}$ be a $\delta$-correct algorithm under which each agent communicates each reward to the central server after it is observed, and let us denote for any $k \in [K]$ , $m \in [M]$ , $\tau_{k,m} := \mathbb{E}_\mu^\mathcal{A} [N_{k,m}(\tau)]$ , where $\tau$ is the stopping time. For any $m \in [M]$ and $k \neq k_m^\star$, it holds that*

$$\sum_n w_{n,m}^2 \left( \frac{1}{\tau_{k,n}} + \frac{1}{\tau_{k_m^\star,n}} \right) \leq \frac{\left( \Delta_{k,m}' \right)^2 / 2}{\log(1/(2.4\delta))} ,$$

*and therefore $\mathbb{E}_\mu \left[ Exp_\mu(\mathcal{A}) \right] \geq T_W^\star(\mu') \log \left( \frac{1}{2.4\delta} \right)$, where*

---

[3]Note that, in a regret setting, an idle agent may exploit its empirical best arm.

$$T_W^\star(\mu) \coloneqq \min_{t \in (\mathbb{R}^+)^{K \times M}} \left\{ \sum_{(k,m) \in [K] \times [M]} t_{k,m} : \forall m, k \neq k_m^\star, \sum_{n \in [M]} w_{n,m}^2 \left( \frac{1}{t_{k,n}} + \frac{1}{t_{k_m^\star,n}} \right) \leq \frac{\left(\Delta_{k,m}'\right)^2}{2} \right\} .$$

The proof, given in Appendix A.1, uses standard change-of-distribution arguments, together with classical results from constrained optimization. Note that, for $M = 1$, we recover the complexity of best arm identification in a Gaussian bandit model [15].

**Computing the complexity term** The optimization problem which defines $T_W^\star(\mu)$ belongs to the family of disciplined convex optimization problems, and can be numerically solved using available solvers, such as CVXPY [1, 9]. We now illustrate, on a small example, the possible reduction in exploration cost that can be obtained by solving weighted collaborative best arm identification instead of $M$ parallel best arm identification problems. We consider $K = M = 2$ and a similarity $S(1,2) = 0.9$ between the two agents, which yields the following normalized weight matrix

$$W = \frac{1}{1.9} \begin{bmatrix} 1 & 0.9 \\ 0.9 & 1 \end{bmatrix} .$$

Considering the following matrix of expected rewards

$$\mu = \begin{bmatrix} 0.9 & 0.8 \\ 0.1 & 0.5 \end{bmatrix} ,$$

for which arm 1 is the local best arm for both agents and is also their best mixed arm, we obtain $T_W^\star(\mu) \approx 28$. However, if each agent solves its own best arm identification problem in isolation (which amounts to using $W = \mathrm{Id}_2$), the resulting exploration cost scales with $T_{\mathrm{Id}_2}^\star(\mu) \approx 101 > 3T_W^\star(\mu)$.

**From lower bounds to algorithms** In single-agent pure exploration tasks, lower bounds are usually guidelines to design optimal algorithms, as they allow to recover an oracle allocation (*i.e.*, the $\arg\min$ for $t \in (\mathbb{R}^+)^{K,M}$ in the definition of $T_W^\star(\mu)$) which algorithms can try to achieve by using some tracking [15, 10, 29]. Yet, these approaches may be computationally expensive, as they solve the optimization problem featured in the lower bound in every round.

In the next section, we will propose an alternative approach for our collaborative setting, which exploits the knowledge of the lower bound within a phased elimination algorithm. This is crucial to maintain a small communication cost and also permit to reduce the computational complexity compared to a pure tracking approach. Our algorithm will rely on a *relaxed* complexity term $\widetilde{T}_W^\star(\mu)$, which is within constant factors of $T_W^\star(\mu)$, as proved in Appendix C.1.

**Lemma 1.** *Introducing the quantity*

$$\widetilde{T}_W^\star(\mu) \coloneqq \min_{t \in (\mathbb{R}^+)^{K \times M}} \left\{ \sum_{(k,m) \in [K] \times [M]} t_{k,m} : \forall m, \forall k \in [K], \sum_{n \in [M]} \frac{w_{n,m}^2}{t_{k,n}} \leq \frac{\left(\Delta_{k,m}'\right)^2}{2} \right\} ,$$

*it holds that* $\widetilde{T}_W^\star(\mu) \leq T_W^\star(\mu) \leq 2\widetilde{T}_W^\star(\mu)$.

Compared to $T_W^\star(\mu)$, a nice feature of $\widetilde{T}_W^\star(\mu)$ is that its constraint set does not depend on the knowledge of $(k_m^\star)_{m \in [M]}$, which will allow us to design algorithms that do not suffer too much from bad empirical guesses for $k_m^\star$ in early phases. We further remark that the computation of $\widetilde{T}_W^\star(\mu)$ and that of its associated oracle allocation (see Definition 1) are slightly easier than for $T_W^\star(\mu)$. Indeed, the optimization problem which defines $\widetilde{T}_W^\star(\mu)$ can be decoupled across arms. Computing for every arm $k \in [K]$ the vector

$$\widetilde{\tau}^k = \underset{\tau^k \in (\mathbb{R}^+)^M}{\mathrm{argmin}} \sum_{(k,m) \in [K] \times [M]} \tau_m^k \text{ s.t. } \forall m, \sum_{n \in [M]} \frac{w_{n,m}^2}{\tau_n^k} \leq \frac{\left(\Delta_{k,m}'\right)^2}{2} ,$$

we obtain the argmin in $\widetilde{T}_W^\star(\mu)$ by setting $(t_{k,m})_{k,m} = (\widetilde{\tau}_m^k)_{k,m}$. The computation of $\widetilde{T}_W^\star(\mu)$ can therefore be done by solving $K$ disciplined optimization problems (*e.g.*, with CVXPY [1, 9]) involving $M$ variables, instead of one optimization problem with $K \times M$ variables.

**Definition 1.** *For any $\Delta \in (\mathbb{R}^+)^{K \times M}$, the oracle $\widetilde{\mathcal{P}}^\star(\Delta)$ is*

$$\underset{(\tau_{k,m})_{k,m} \in (\mathbb{R}^+)^{K \times M}}{\arg\min} \sum_{k,m} \tau_{k,m} \; s.t. \forall m \in [M], \forall k \in [K], \sum_{n \in [M]} \frac{w_{n,m}^2}{\tau_{k,n}} \leq \frac{(\Delta_{k,m})^2}{2} \; .$$

With this notation, observe that $\widetilde{T}^\star(\mu') = \sum_{k,m} \tau_{k,m}$ , where $(\tau_{k,m})_{k,m} \in \widetilde{\mathcal{P}}^\star(\Delta')$. The following lemma will be useful to compare values from different oracle problems. The full lemma along with its proof is available in Appendix C.1.

**Lemma 2.** *Consider $\Delta$ , $\Delta' \in (\mathbb{R}^+)^{K \times M}$, such that $\tau \in \widetilde{\mathcal{P}}^\star(\Delta)$ and $\tau' \in \widetilde{\mathcal{P}}^\star(\Delta')$. Moreover, assume that there is a positive constant $\beta$ such that: $\forall k, m, \Delta'_{k,m} \leq \beta \Delta_{k,m}$. Then*

$$\frac{1}{\beta^2} \sum_{k,m} \tau_{k,m} \leq \sum_{k,m} \tau'_{k,m} \; .$$

## 4 A Near-Optimal Algorithm For Best Arm Identification

We now introduce an algorithm for collaborative best-arm identification, called W-CPE-BAI for Weighted Collaborative Phased Elimination, stated as Algorithm 1. To present an analysis of this algorithm, we assume that, for any $k, m \in [K] \times [M]$ , $\mu_{k,m} \in [0,1]$ , and that local rewards $(X_{k,m}(t))_{k,m,t}$ are 1-sub-Gaussian.

W-CPE-BAI proceeds in phases, indexed by $r$ . In phase $r$ , we let $B_m(r)$ be the set of active arms for agent $m$ , and $B(r) = \cup_m B_m(r)$ be the set of arms that are active for at least one agent. The algorithm maintains proxies for the gaps $(\widetilde{\Delta}_{k,m}(r))_{k \in [K], m \in [K]}$ that are halved at the end of each phase for arms that remain active. At the beginning of each round, the oracle allocation $t(r)$, with respect to the proxy gaps, is computed, as well as the number of new samples $d_{k,m}(r)$ that player $m$ should get from arm $k$ in phase $r$ . $d_{k,m}(r)$ is defined such that the total number of selections of arm $k$ by agent $m$ becomes close to (a quantity slightly larger than) $t_{k,m}(r) \log(1/\delta)$ . We observe that any arm $k \notin B(r)$ will not get any new samples in phase $r$ , as the proxy gaps $(\widetilde{\Delta}_{k,n}(r))_n$ are identical to those in the previous phase ; therefore $t_{k,n}(r) = t_{k,n}(r-1)$ . In contrast to prior works, where the allocation in each round only depends on the identity of the surviving arms and the round index [13, 10], in W-CPE-BAI it also depends on when the arms have been eliminated (which condition the value of their frozen proxy gaps).

After each agent $m$ samples arm $k$ $d_{k,m}(r)$ times, they send their local means $\hat{\mu}_{k,m}(r)$ to the central server, which computes the mixed mean estimates $\hat{\mu}'_{k,n}(r) := \sum_{m=1}^M w_{n,m}\hat{\mu}_{k,m}(r)$ . The active sets $(B_m(r))_{m \in [M]}$ of all agents are then updated by removing arms whose mixed means are too small. As in several prior works [22, 30], we rely on confidence intervals to perform these eliminations. However, constructing confidence intervals on the mixed means –which are linear combinations of the local means– under our adaptive sampling rule is more challenging than when the number of samples from an active arm in phase $r$ is fixed in advance –which is the case for instance in the algorithm in [30]. The width of our confidence intervals scales with the following quantity

**Definition 2.** *For any $k, m$, and round $r \geq 0$, we define*

$$\Omega_{k,m}(r) := \sqrt{\beta_\delta(n_{k,\cdot}(r)) \sum_{n=1}^M \frac{w_{n,m}^2}{n_{k,n}(r)}} \; ,$$

*where $n_{k,m}(r)$ is the number of times arm $k$ was selected by agent $m$ by the end of phase $r$ (included), and $N \mapsto \beta^\delta(N)$ is a threshold function defined for any $N \in (\mathbb{R}^+)^M$ .*

Leveraging some recent time-uniform concentration inequalities [23], we exhibit below a choice of threshold that yields valid confidence intervals on the mixed means (by "projecting" confidence intervals that can be obtained on local means, see Proposition 24 in [23]). The fact that the confidence interval depends on the random number of past draws (and not just the index of the round) leads to some non-trivial complication in the analysis, with the introduction of quantities $(d_{k,m})_{k,m}$. The proof is given in Appendix C.2, where we also provide an explicit expression of the function $g_M$ .

**Algorithm 1** Weighted Collaborative Phased Elimination for Best Arm Identification (W-CPE-BAI)

**Input:** $\delta \in (0,1)$, $M$ agents, $K$ arms, weights matrix $W$
Initialize $r \leftarrow 0$, $\forall k, m, \widetilde{\Delta}_{k,m}(0) \leftarrow 1$, $n_{k,m}(0) \leftarrow 1$, $\forall m, B_m(0) \leftarrow [K]$
Draw each arm $k$ by each agent $m$ once
**repeat**
  # Central server
  $B(r) \leftarrow \bigcup_{m \in [M]} B_m(r)$
  Compute $t(r) \leftarrow \widetilde{\mathcal{P}}^\star \left( \left( \sqrt{2}\widetilde{\Delta}_{k,m}(r) \right)_{k,m} \right)$
  For any $k \in [K]$, compute

$$(d_{k,m}(r))_{m \in [M]} \leftarrow \arg\min_{d \in \mathbb{N}^M} \sum_m d_m \text{ s.t. } \forall m \in [M], \frac{n_{k,m}(r-1) + d_m}{\beta_\delta(n_{k,\cdot}(r-1)+d)} \geq t_{k,m}(r)$$

  Send to each agent $m$ $(d_{k,m}(r))_{k,m}$ and $d_{\max} := \max_{n \in [M]} \sum_{k \in [K]} d_{k,n}(r)$

  # Agent $m$
  Sample arm $k \in B(r)$ $d_{k,m}(r)$ times, so that $n_{k,m}(r) = n_{k,m}(r-1) + d_{k,m}(r)$
  Remain idle for $d_{\max} - \sum_{k \in [K]} d_{k,m}(r)$ rounds
  Send to the server empirical mean $\hat{\mu}_{k,m}(r) := \sum_{s \leq n_{k,m}(r)} X_{k,m}(s)/n_{k,m}(r)$ for any $k \in [K]$

  # Central server
  Compute the empirical mixed means $(\hat{\mu}'_{k,m}(r))_{k,m}$ based on $(\hat{\mu}_{k,m}(r))_{k,m}$ and $W$
  *// Update set of candidate best arms for each user*
  **for** $m = 1$ **to** $M$ **do**

$$B_m(r+1) \leftarrow \left\{ k \in B_m(r) \mid \hat{\mu}'_{k,m}(r) + \Omega_{k,m}(r) \geq \max_{j \in B_m(r)} \left( \hat{\mu}'_{j,m}(r) - \Omega_{j,m}(r) \right) \right\}$$

  **end for**
  *// Update the gap estimates*
  For all $k, m$, $\widetilde{\Delta}_{k,m}(r+1) \leftarrow \widetilde{\Delta}_{k,m}(r) \times (1/2)^{\mathbb{1}(k \in B_m(r+1) \wedge |B_m(r+1)| > 1)}$
  $r \leftarrow r + 1$
**until** $\forall m \in [M], |B_m(r)| \leq 1$
**Output:** $\{k \in B_m(r) : m \in [M]\}$

---

**Lemma 3.** *Let us define*

$$\beta_\delta(N) := 2 \left( g_M \left( \frac{\delta}{KM} \right) + 2 \sum_{m=1}^M \ln(4 + \ln(N_m)) \right),$$

*for any $N \in (\mathbb{N}^\star)^M$, where $g_M$ is some non-explicit function, defined in [23], that satisfies $g_M(\delta) \simeq \log\left(\frac{1}{\delta}\right) + M \log\log\left(\frac{1}{\delta}\right)$. Then the good event*

$$\mathcal{E} := \left\{ \forall r \in \mathbb{N}, \forall m, \forall k, \left| \hat{\mu}'_{k,m}(r) - \mu'_{k,m} \right| \leq \Omega_{k,m}(r) \right\}$$

*holds with probability larger than $1 - \delta$.*

From this lemma, it easily follows that W-CPE-BAI is $\delta$-correct for the above choice of threshold function, as, for any agent $m$, no good arm $k_m^\star$ can ever be eliminated from $B_m(r)$ at round $r$. The fact that the sample complexity of W-CPE-BAI scales with $\widetilde{T}^\star(\mu')$ on the good event $\mathcal{E}$ comes from the interplay between the expression of $\Omega_{k,m}(r)$ (which, up to the threshold function, is exactly one of the constraints featured in the lower bound) together with the definition of the allocation $t(r)$, which leads to the following crucial result in our analysis

**Lemma 4.** *On $\mathcal{E}$, $\forall k, m, r \geq 0$, $\Omega_{k,m}(r) \leq \widetilde{\Delta}_{k,m}(r)$.*

*Proof.* For any round $r$ and arm $k$, by Algorithm 1 and the definition of oracle $t_{k,\cdot}(r)$, for any agent $m$,

$$\Omega_{k,m}(r) \;=\; \sqrt{\sum_n w_{n,m}^2 \frac{\beta_\delta(n_{k,\cdot}(r-1)+d_{k,\cdot})}{n_{k,n}(r-1)+d_{k,n}(r)}} \;\le\; \begin{cases} \sqrt{\sum_n \frac{w_{n,m}^2}{t_{k,n}(r)}} \le \widetilde{\Delta}_{k,m}(r) & \text{if } k \in B(r)\,, \\[2mm] \sqrt{\sum_n \frac{w_{n,m}^2}{t_{k,n}(r_k')}} \le \widetilde{\Delta}_{k,m}(r_k') = \widetilde{\Delta}_{k,m}(r) & \text{otherwise}\,, \end{cases}$$

where when $k \notin B(r)$, $r_k' := \sup\{r' \ge 0 : k \in B(r')\}$, and we use the fact that $d_{k,m}(r) = 0$ when $k \notin B(r)$. $\qquad\square$

We did not put much emphasis on the way communications are performed between the agents and the central server, as several choices are possible. The important part is that the server receives all values of the local means $(\hat{\mu}_{k,m}(r))_{k,m}$ at the end of round $r$. Our suggestion is that the central server maintains the sets $(B_m(r))_{m\in[M]}$, calls the oracle, and sends to all agents their values of $(d_{k,m}(r+1))_{k,m}$ at the end of each phase $r$. In any case, the number of communication rounds in our definition will be equal to the number of phases used by W-CPE-BAI. All in all, we prove

**Theorem 2.** *With probability $1 - \delta$, W-CPE-BAI outputs the optimal arm for each agent with an exploration cost at most*

$$32\widetilde{T}_W^\star(\mu)\log_2(8/\Delta'_{\min})\log\left(\frac{1}{\delta}\right) + o_\delta\left(\log\left(\frac{1}{\delta}\right)\right)\,,$$

*and at most $\lceil \log_2(8/\Delta'_{\min})\rceil$ communication rounds, where $\Delta'_{\min} := \min_{k\in[K],m\in[M]} \Delta'_{k,m}$.*

**Proof sketch** The detailed proof is given in Appendix B, where we also provide an explicit upper bound on the exploration cost. We let $R$ denote the (random) number of phases used by the algorithm before stopping. On the good event $\mathcal{E}$, we can prove that the algorithm never eliminates $k_m^\star$ therefore $R := \max_m \max_{k\neq k_m^\star} R_{k,m}$ where $R_{k,m}$ is the last phase in which $k \in B_m(r)$. Using Lemma 4, we can easily establish that

$$R_{k,m} \le r_{k,m} := \min\left\{r \ge 0 : 4 \times 2^{-r} < \Delta'_{k,m}\right\}$$

which satisfies $r_{k,m} \le \log_2(8/\Delta'_{k,m})$. This yields $R \le \log_2(8/\Delta'_{\min})$, and further permits to prove that the proxy gaps can be lower bounded by the true gaps:

$$\forall r \le R, \forall k \in [K], \forall m \in [M], \; \widetilde{\Delta}_{k,m}(r) \ge \frac{1}{8}\Delta'_{k,m}\,.$$

See Corollary 2 in Appendix B. Using the monotonicity properties of the oracle that are stated in Lemma 2, we can then establish that the allocation $t(r)$ computed from the proxy gaps in the algorithm satisfies

$$\forall r \le R, \quad \sum_{k\in[K],m\in[M]} t_{k,m}(r) \le 32\widetilde{T}_W^\star(\mu). \tag{1}$$

To upper bound the exploration cost, the next step is to relate $n_{k,m}(R)$ to the oracle allocations. To do so, we observe that if $R'_{k,m}$ is the last round before $R$ such that $d_{k,m}(r) \neq 0$ (i.e. the last round in which arm $k$ is actually sampled by agent $m$; then $n_{k,m}(R) = n_{k,m}(R'_{k,m})$), we have by definition of the $(d_{k,m}(r))_{k,m,r}$ that

$$n_{k,m}(R'_{k,m}) \;\le\; t_{k,m}(R'_{k,m})\beta_\delta(n_{k,\cdot}(R'_{k,m})) + 1 \le t_{k,m}(R'_{k,m})\beta_\delta(n_{k,\cdot}(R)) + 1\,.$$

See Lemma 12 in Appendix B. We can then upper bound $\tau := \sum_{k,m} n_{k,m}(R)$ as follows

$$\tau \;\le\; \sum_{k,m} t_{k,m}(r'_{k,m})\beta_\delta(n_{k,\cdot}(R)) + KM \le \sum_{k,m}\sum_{r\le R} t_{k,m}(r)\beta^*(\tau) + KM\,,$$

$$\text{and } \tau \;\le\; R \times 32\widetilde{T}_W^\star(\mu)\beta^*(\tau) + KM\,, \text{ where we use (1) and introduce} \tag{2}$$

$$\beta^*(\tau) := 2\left(g_M\left(\frac{\delta}{KM}\right) + 2M\ln\left(4 + \ln(\tau)\right)\right)\,.$$

The end of the proof consists in using the known upper bound on $R$, and finding an upper bound for the largest $\tau$ satisfying the inequality in (2). $\qquad\square$

**Discussion**    Theorem 2 proves that W-CPE-BAI is matching the exploration lower bound of Theorem 1 in a regime where $\delta$ is small, up to multiplicative constants, including a logarithmic term in $1/\Delta'_{\min}$. It achieves this using only $\lceil \log_2 (8/\Delta'_{\min}) \rceil$ communication rounds. We note that a similar extra multiplicative logarithmic factor is present in the analysis of near-optimal phased algorithms in other contexts [13, 10]. Such a quantity appears as an upper bound on the number of phases, and may be a price to pay for the phased structure.

**On the communication cost**    We argue that the communication cost of W-CPE-BAI is actually of the same order of magnitude as that featured in some related work. In [30], which is the closest setting to our framework, the equivalent number of communication rounds $p$ needed to solve the regret minimization problem is upper bounded by $\mathcal{O}\left(2\log_2\left(8/(\sqrt{M}\Delta'_{\min})\right)\right)$. In the setting of collaborative learning –where $M$ agents face the same set of arm distributions and $W = Id$– [18] in their Theorem $4.1$ prove that an improvement of multiplicative factor $1/M$ on the exploration cost for a traditional best arm identification algorithm can be reached by using at most $\lceil \log_2(1/\Delta_{\min}) \rceil$ communication rounds, where $\Delta_{\min}$ is the gap between the best and second best arms.

**Experimental validation**    We propose in Appendix E an empirical evaluation of W-CPE-BAI for the weight matrix $w_{m,n} = \alpha\mathbb{1}(n = m) + \frac{1-\alpha}{M}$ which corresponds to the setting studied by [30]. In this particular case, we propose as a baseline a counterpart of the regret algorithm of [30] which we call PF-UCB-BAI. Our experiments on a synthetic instance show that W-CPE-BAI and PF-UCB-BAI have similar performances in terms of exploration cost and that W-CPE-BAI becomes better when the level of personalization $\alpha$ is smaller than $0.5$. Moreover, W-CPE-BAI uses less rounds of communication than PF-UCB-BAI for all values of $\alpha$. Finaly, the near-optimality of W-CPE-BAI is empirically observed when compared to an oracle algorithm which has access to the true gaps. We refer the reader to Appendix E for further details on the optimization libraries that were used.

**Remark 1.** *The analysis and the structure of Algorithm 1 have the potential to be extended to other pure exploration problems, with similar guarantees. In Appendix D, we illustrate this claim by tackling Top-$N$ identification.*

## 5    Regret Lower Bound

In contrast to the BAI setting, [30] considered the objective of minimizing the regret

$$\mathcal{R}_\mu(T) \ := \ \mathbb{E}\left[ \sum_{m=1}^M \sum_{t=1}^T \left( \mu'_{k^\star_m,m} - \mu_{\pi_m(t),m} \right) \right].$$

They provided a conjecture on the lower bound on regret in personalized federated learning [see, 30, Conjecture 1]. As mentioned in introduction, their reward model is a special case of ours with weights $w_{m,m} = \alpha + \frac{1-\alpha}{M}$ and $w_{n,m} = \frac{1-\alpha}{M}$ for $n \neq m$. In this section, we prove a regret lower bound with general weights that proves in particular Conjecture 1 in [30].

We prove the following result, for an algorithm that selects in each round exactly one arm for each agent, and all agents communicate after each round. This lower bound holds for Gaussian arms with variance $\sigma^2 = 1$.

**Theorem 3.** *Any uniformly efficient algorithm[4] in which all agents communicate after each round satisfies*

$$\liminf_{T\to\infty} \frac{\mathcal{R}(T)}{\log(T)} \geq C^\star_W(\mu) \,,$$

*where*

$$C^\star_W(\mu) := \min_{c=(c_{k,m})_{k,m:k^\star_m \neq k}} \left\{ \sum_{k=1}^K \sum_{m:k^\star_m \neq k} c_{k,m}\Delta'_{k,m} \ : \ \forall k \in [K], \forall m \in [M], \sum_{n:k^\star_n \neq k} \frac{w^2_{n,m}}{c_{k,n}} \leq \frac{(\Delta'_{k,m})^2}{2} \right\} \,.$$

*We recall that for any agent $m$, $\Delta'_{k^\star_m,m} := \min_{k' \neq k^\star_m} \Delta'_{k',m}$.*

---

[4]A uniformly efficient algorithm satisfies $\mathcal{R}_\mu(T) = o(T^\gamma)$ for any $\gamma \in (0,1)$ and any possible instance $\mu$.

Theorem 3 may be viewed as an extension of the lower bound of [25] to the collaborative setting. Its proof, given in Appendix A.2, uses classical ingredients for regret lower bounds in (single agent) structured bandit models [7, 17].

The generic approaches proposed in [7, 8] for optimal regret minimization in structured bandits could therefore be useful. However, to turn them into a reasonable algorithm for the collaborative setting, we would need to preserve the phased elimination structure. Analogous to the relaxation in the BAI setting, we can also define a relaxed complexity term $\widetilde{C}_M(\mu)$, which does not require the knowledge of the best arms, and is within constant factors of $C(\mu)$ by the following Lemma 5 (which proof is given in Appendix C.3).

**Lemma 5.** *Introducing the quantity*

$$\widetilde{C}_W^\star(\mu) := \min_{c\in(\mathbb{R}^+)^{K\times M}} \left\{ \sum_{k=1}^{K} \sum_{m=1}^{M} c_{k,m}\Delta'_{k,m} \; s.t. \; \forall k\in[K], \forall m\in[M], \sum_{n=1}^{M} \frac{w_{n,m}^2}{c_{k,n}} \leq \frac{(\Delta'_{k,m})^2}{2} \right\},$$

*it holds that* $C_W^\star(\mu) \leq \widetilde{C}_W^\star(\mu) \leq 4C_W^\star(\mu)$ .

The constrained optimization problems governing the regret lower bound are subtly different from those in the BAI setting. In regret minimization, unknown gaps $(\Delta'_{k,m})_{k,m}$ appear both in the constraints and in the objective. In contrast, in BAI, $(\Delta'_{k,m})_{k,m}$ only appear in the constraints. Due to this difference, designing a similar algorithm for regret minimization leads to an extra multiplicative $\mathcal{O}(1/\Delta'_{\min})$ factor in the upper bound. A very similar issue exists in a different structured bandit problem, bandits on graphs with side observations, where arms are connected through a graph with unweighted edges, and the agent receives the reward associated with the selected arm *and its neighbors* at a given round. In [4, Problem P1], authors describe a constrained linear optimization problem which governs the regret lower bound, and face the same issue of scaling in $\mathcal{O}(1/\Delta'_{\min})$. Dealing with this problem would be an interesting subsequent work.

## 6 Conclusion

This paper introduced a general framework for collaborative learning in multi-armed bandits. Our work presents two novel lower bounds: one on the exploration cost for pure exploration, and another on cumulative regret. The latter permits to prove a prior conjecture on the topic. Moreover, we propose a phased elimination algorithm for fixed-confidence best arm identification. This algorithm tracks the optimal allocation from the pure exploration lower bound through a novel approach solving a relaxed optimization problem linked to the lower bound. The exploration cost of this algorithm is matching the lower bound up to logarithmic factors. This strategy can be extended to other pure exploration problems, such as Top-$N$ identification, as shown in Appendix D.

As mentioned in introduction, our collaborative setting was motivated by the design of collaborative adaptive clinical trials for personalized drug recommendations, where several patient subpopulations (for instance, representing several subtypes of cancer) are considered and sequentially treated. However, in practice, especially when dealing with patient data, disclosing the mean response values to the central server should be handled with care to preserve the anonymity of the patients. A possible solution to overcome this problem would be to carefully combine our algorithm with a data privacy-preserving method, for instance by adding some noise to the data [11].

Our code and run traces are available in an open-source repository. [5]

## Acknowledgments and Disclosure of Funding

Clémence Réda was supported by the French Ministry of Higher Education and Research [ENS.X19RDTME-SACLAY19-22]. The authors acknweldge the support of the French National Research Agency under the project [ANR-19-CE23-0026-04] (BOLD).

---

[5] `https://github.com/clreda/near-optimal-federated`

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
