# A Proof of the Lower Bounds

## A.1 Lower Bound on the Exploration Cost: Proof of Theorem 1

Let us denote, for any model $\mu \in \mathbb{R}^{K \times M}$ and agent $m$ , $k_m^\star := \arg\max_{k \in [K]} \mu_{k,m}'$ (which is assumed unique). Define the set of alternative models in $\mathbb{R}^{K \times M}$ with respect to $\mu$ :

$$\mathrm{Alt}(\mu) := \left\{ \lambda : \exists m, \exists k \neq k_m^\star : \lambda_{k,m}' > \lambda_{k_m^\star,m}' \right\} ,$$

where $\lambda_{k,m}' := \sum_{n \in [M]} w_{n,m} \lambda_{k,n}$ for any arm $k$ and agent $m$ . Assume that stopping time $\tau$ is almost surely finite under $\mu$ for algorithm $\mathcal{A}$. Let event $\mathcal{E}_\mu := \left\{ \exists m : \hat{k}_m \neq k_m^\star \right\}$ . As algorithm $\mathcal{A}$ is $\delta$-correct, it holds that $\mathbb{P}_\mu(\mathcal{E}_\mu) \leq \delta$ and $\mathbb{P}_\lambda(\mathcal{E}_\mu) \geq 1 - \delta$ for all $\lambda \in \mathrm{Alt}(\mu)$ . As this event belongs to the filtration generated by all past observations up to the final stopping time $\tau$, using Theorem 1 from [15] and $\delta \leq 1/2$ , it holds that

$$\frac{1}{2} \sum_{k,m} \tau_{k,m} (\mu_{k,m} - \lambda_{k,m})^2 \geq \log\left(\frac{1}{2.4\delta}\right) . \tag{3}$$

We first prove that, for any $k, m$ , $\tau_{k,m} > 0$ . Indeed, if $w_{m,m} \neq 0$ , it is possible to pick $\lambda$ that only differs from $\mu$ by the entry $\lambda_{k,m}$ , in such a way that arm $k$ becomes optimal (or sub-optimal) for user $m$ . From Equation (3), we get $\frac{1}{2} \tau_{k,m}(\mu_{k,m} - \lambda_{k,m})^2 > 0$ and the conclusion follows.

We now fix agent $m$ and $k \neq k_m^\star$ , and try to find a more informative alternative model $\lambda$ . We look for it in a family of alternative models under which only arms $k$ and $k_m^\star$ are modified, for *all* agents, in order to make arm $k$ optimal for agent $m$ . Given two nonnegative sequences $(\delta_n)_{n \in [M]}$ and $(\delta_n')_{n \in [M]}$ , we define $\lambda = (\lambda_{k',n})_{k'}$ such that

$$\left\{ \begin{array}{rcl} \lambda_{k',n} & = & \mu_{k',n} \text{ if } k' \notin \{k, k_m^\star\} , \\ \lambda_{k,n} & = & \mu_{k,n} + \delta_n , \\ \lambda_{k_m^\star,n} & = & \mu_{k_m^\star,n} - \delta_n' , \end{array} \right.$$

that satisfies

$$\sum_{n \in [M]} w_{n,m} (\delta_n + \delta_n') \geq \Delta_{k,m}' . \tag{4}$$

Now arm $k$ is optimal for agent $m$. From Equation (3),

$$\sum_n \tau_{k,n} \frac{\delta_n^2}{2} + \sum_n \tau_{k_m^\star,n} \frac{(\delta_n')^2}{2} \geq \log\left(\frac{1}{2.4\delta}\right) .$$

Hence, it holds that

$$\inf_{\delta,\delta':(4)\text{ holds}} \left[ \sum_n \tau_{k,n} \frac{\delta_n^2}{2} + \sum_n \tau_{k_m^\star,n} \frac{(\delta_n')^2}{2} \right] .$$

The infimum can be computed in closed form using constraint optimization. Introducing a Lagrange multiplier $\lambda$, from the KKT conditions, we get that, for any agent $n$ ,

$$\begin{array}{rcl} \tau_{k,n} \delta_n - \lambda w_{n,m} & = & 0 , \\ \tau_{k_m^\star,n} \delta_n' - \lambda w_{n,m} & = & 0 , \\ \lambda \left( \sum_{n'} w_{n',m} (\delta_{n'} + \delta_{n'}') - \Delta_{k,m}' \right) & = & 0 . \end{array}$$

Using furthermore that $\tau_{k,n}$ and $\tau_{k_m^\star,n}$ are positive,

$$\delta_n = \frac{\Delta_{k,m}' w_{n,m}/\tau_{k,n}}{\sum_{n' \in [M]} w_{n',m}^2 \left(\frac{1}{\tau_{k,n'}} + \frac{1}{\tau_{k_m^\star,n'}}\right)} \quad \text{and} \quad \delta_n' = \frac{\Delta_{k,m}' w_{n,m}/\tau_{k_m^\star,n}}{\sum_{n' \in [M]} w_{n',m}^2 \left(\frac{1}{\tau_{k,n'}} + \frac{1}{\tau_{k_m^\star,n'}}\right)} .$$

The conclusion follows by plugging these expressions to get the expression of the infimum.

## A.2 Regret Lower Bound: Proof of Theorem 3

Given a bandit instance $\mu$, we can consider two sets of possible changes of distributions: changes that are allowed to change the distributions of optimal arms $(k_m^*)_{m \in [M]}$, and those that cannot

$$\mathrm{Alt}(\mu) \quad := \quad \left\{ \lambda = (\lambda_{k,m})_{k,m} \in \mathbb{R}^{K \times M} : \exists m \in [M], k \neq k_m^* : \sum_n \lambda_{k,m} w_{n,m} > \sum_n \lambda_{k_m^*,n} w_{n,m} \right\} ,$$

$$B(\mu) \quad := \quad \mathrm{Alt}(\mu) \bigcap \left\{ \lambda = (\lambda_{k,m})_{k,m} \in \mathbb{R}^{K \times M} : \forall m \in [M], \lambda_{k_m^*,m} = \mu_{k_m^*,m} \right\} .$$

For cumulative regret, the change-of-distribution lemma becomes an asymptotic result, stated below

**Lemma 6.** *Fix $\mu \in \mathbb{R}^{K \times M}$, and let us consider $\lambda \in \mathrm{Alt}(\mu)$. Then, for any $\varepsilon > 0$, there exists $T_0 = T_0(\mu, \lambda, \varepsilon)$ , such that, for any $T \geq T_0$*

$$\sum_{m,k} \mathbb{E}_\mu[N_{k,m}(T)] \frac{(\mu_{k,m} - \lambda_{k,m})^2}{2} \geq (1 - \varepsilon) \log(T) .$$

*Proof.* The proof uses a change-of-distribution, following a technique proposed by [16]. Using the data processing inequality, and letting $\mathcal{H}_T$ be the observations available to the central server (which sees all local rewards under our assumptions), we have that

$$\mathrm{KL}\left(\mathbb{P}_\mu^{\mathcal{H}_T}, \mathbb{P}_\lambda^{\mathcal{H}_T}\right) \geq \mathrm{kl}\left(\mathbb{P}_\mu(\mathcal{E}_T), \mathbb{P}_\lambda(\mathcal{E}_T)\right) ,$$

where KL is the Kullback-Leibler divergence and $\mathbb{P}_\mu^{\mathcal{H}_T}$ is the distribution of the observation under the bandit model $\mu$ , and $\mathcal{E}_T$ is any event. Using that

$$\mathrm{KL}\left(\mathbb{P}_\mu^{\mathcal{H}_T}, \mathbb{P}_\lambda^{\mathcal{H}_T}\right) = \sum_{k,m} \mathbb{E}_\mu[N_{k,m}(T)] \frac{(\mu_{k,m} - \lambda_{k,m})^2}{2}$$

together with the lower bound $\mathrm{kl}(p, q) \geq (1 - p) \log(1/(1 - q)) - \log(2)$, where kl is the binary relative entropy and for any distributions $p, q$, yields

$$\sum_{k,m} \mathbb{E}_\mu[N_{k,m}(T)] \frac{(\mu_{k,m} - \lambda_{k,m})^2}{2} \geq (1 - \mathbb{P}_\mu(\mathcal{E}_T)) \log\left(\frac{1}{\mathbb{P}_\lambda(\overline{\mathcal{E}_T})}\right) - \log(2) .$$

We now pick the event

$$\mathcal{E}_T = \left\{ N_{k_m^*, T}(T) \leq \frac{T}{2} \right\} ,$$

and use that $\mathcal{E}_T$ is very unlikely under $\mu$ as for any $\gamma \in (0, 1)$ ,

$$\mathbb{P}_\mu(\mathcal{E}_T) = \mathbb{P}_\mu\left( \sum_{k \neq k_m^*} N_{k,m}(T) \geq \frac{T}{2} \right) = \frac{2 \sum_{k \neq k_m^*} \mathbb{E}_\mu[N_{k,m}(T)]}{T} = \frac{o_{T \to \infty}(T^\gamma)}{T} ,$$

and very likely under any $\lambda$ for which $k_m^*$ is suboptimal as, for any $\gamma \in (0, 1)$ ,

$$\mathbb{P}_\lambda(\overline{\mathcal{E}_T}) = \mathbb{P}_\lambda\left( N_{k_m^*, T}(T) > \frac{T}{2} \right) = \frac{2 \mathbb{E}_\lambda[N_{k_m^*, m}(T)]}{T} = \frac{o_{T \to \infty}(T^\gamma)}{T} ,$$

where we exploit the fact that the algorithm is uniformly efficient (its regret and therefore its number of sub-optimal draws is $o(T^\gamma)$ under any bandit model). The conclusion follows from some elementary real analysis to prove that the right hand side of the inequality is larger than $(1 - \varepsilon) \log(T)$ for $T$ large enough (how larger depends in a complex way of $\mu, \lambda, \varepsilon$ and the algorithm). $\square$

At this point, we would really like to select the alternative model $\lambda$ that leads to the tightest inequality in Lemma 6. For example, using Lemma 7 below, we find that the best way to make an arm $k \neq k_m^*$ better than $k_m^*$ consists in choosing

$$\lambda_{k,n}(T) = \mu_{k,n} + \frac{\Delta_{k,m} w_{n,m} / \mathbb{E}_\mu[N_{k,m}(T)]}{\sum_{m' \in [M]} \frac{w_{n',m}^2}{\mathbb{E}_\mu[N_{k,n'}(T)]}} ,$$

for any $n \in [M]$ and $\lambda_{k',n}(T) = \mu_{k',n}$ for any arm $k' \neq k$. However, this choice of alternative $\lambda$ depends on $T$, hence we cannot apply Lemma 6, which is asymptotic in $T$ and holds for a fixed $\lambda$. We have to be careful to be able to exchange the $\liminf$ over $T$ and the infimum over alternatives in the constraints, and we will be able to do so only for changes of measures that are restricted to $B(\mu)$.

We first assume that $\liminf_{T\to\infty} \frac{\mathcal{R}(T)}{\log(T)}$ is finite, and call its value $\ell(\mu)$ (this is fine as otherwise any lower bound trivially holds). By definition of the $\liminf$, there exists a sequence $(T_i)_{i\in\mathbb{N}}$ such that

$$\liminf_{T\to\infty} \frac{\mathcal{R}(T)}{\log(T)} = \lim_{i\to\infty} \sum_{m\in[M], k\neq k_m^\star} \Delta'_{k,m} \frac{\mathbb{E}_\mu[N_{k,m}(T_i)]}{\log(T_i)} = \ell(\mu) \, .$$

The fact that this sequence has a limit, and that the gaps are positive, implies that each sequence $(\mathbb{E}_\mu[N_{k,m}(T_i)]/\log(T_i))_{i\in\mathbb{N}}$ is bounded. Therefore, there must exist a subsequence, that we denote $(T'_i)_{i\in\mathbb{N}}$ of $(T_i)_{i\in\mathbb{N}}$ such that

$$\forall m \in [M], k \neq k_m^\star, \quad \lim_{i\to\infty} \frac{\mathbb{E}_\mu[N_{k,m}(T'_i)]}{\log(T'_i)} = c_{k,m} \, ,$$

for some real values $(c_{k,m})_{k\in[K], m\in[M]}$, and, in particular,

$$\ell(\mu) = \sum_{m\in[M], k\neq k_m^\star} \Delta'_{k,m} c_{k,m} \, .$$

Now, it follows from Lemma 6 that, for any $\lambda \in \mathrm{Alt}(\mu)$,

$$\liminf_{T\to\infty} \sum_{k,m} \frac{\mathbb{E}_\mu[N_{k,m}(T)]}{\log(T)} \frac{(\mu_{k,m} - \lambda_{k,m})^2}{2} \geq 1 \, .$$

But we have no idea about the behavior of the sequence $(\mathbb{E}_\mu[N_{k,m}(T)]/\log(T))_{T\in\mathbb{N}}$ for $k = k_{m^\star}$, this is why we have to consider only $\lambda \in B(\mu)$, for which we deduce that

$$\lim_{i\to\infty} \sum_{m\in[M], k\neq k_m^\star} \frac{\mathbb{E}_\mu[N_{k,m}(T'_i)]}{\log(T'_i)} \frac{(\mu_{k,m} - \lambda_{k,m})^2}{2} \quad \geq \quad 1 \, ,$$

$$\sum_{m\in[M], k\neq k_m^\star} c_{k,m} \frac{(\mu_{k,m} - \lambda_{k,m})^2}{2} \quad \geq \quad 1 \, .$$

(the $\liminf$ being the lowest value of the limit of any subsequence). This proves that

$$\ell(\mu) \geq \min_c \sum_{m\in[M], k\neq k_m^\star} \Delta'_{k,m} c_{k,m} \, ,$$

under the constraints that $c = (c_{k,m})_{k,m:k\neq k_m^\star}$ belongs to the constraint set

$$\mathcal{C} = \left\{ (c_{k,m})_{k,m:k\neq k_m^\star} : \forall \lambda \in B(\mu), \sum_{m\in[M], k\neq k_m^\star} c_{k,m} \frac{(\mu_{k,m} - \lambda_{k,m})^2}{2} \geq 1 \right\} \, .$$

We now establish that

$$\mathcal{C} \subseteq \bigcup_{k,m} \left\{ (c_{k,n})_{k,m:k\neq k_n^\star} : \sum_{n:k\neq k_n^\star} \frac{w_{n,m}^2}{c_{k,n}} \leq \frac{(\Delta'_{k,m})^2}{2} \right\} \, , \tag{5}$$

by selecting some well chosen elements in $B(\mu)$.

First, for every $(m,k) \in [M] \times [K]$ such that $k \neq k_m^\star$, for every $\delta = (\delta_n)_{n:k_n^\star \neq k}$, we define an instance $\lambda^\delta$ by $\lambda^\delta_{k,n} = \mu_{k,n} + \delta_n$ for any $n \in [M]$ such that $k \neq k_n^\star$, and $\lambda^\delta_{k,n} = \mu_{k,n}$ otherwise. We observe that $\lambda^\delta$ belongs to $B(\mu)$ if

$$\sum_{n\in[M]:k\neq k_n^\star} w_{n,m} \delta_n > \Delta'_{k,m} \, ,$$

as this leads to $\sum_{n\in[M]} w_{n,m} \lambda_{k,n} > \sum_{n\in[M]} w_{n,m} \lambda_{k_m^\star,n}$, and arm $k_m^\star$ being sub-optimal in $\lambda^\delta$. For all $c \in \mathcal{C}$,

$$\min_{\delta: \sum_{n:k\neq k_n^\star} w_{n,m}\delta_n \geq \Delta'_{k,m}} \sum_{n\in[M]:k_n^\star \neq k} c_{k,n} \frac{\delta_n^2}{2} \geq 1 \, .$$

From Lemma 7, this leads to

$$\sum_{n\in[M]:k_n^\star\neq k} \frac{w_{n,m}^2}{c_{k,n}} \leq \frac{(\Delta_{k,m}')^2}{2} \ .$$

Now we consider $(m,k) \in [M] \times [K]$, such that $k = k_m^\star$. In this case, we define an instance $\lambda^\delta$ by $\lambda_{k_m^\star,n}^\delta = \mu_{k_m^\star,n} - \delta_n$ for any $n \in [M]$ such that $k_m^\star \neq k_n^\star$, and $\lambda_{k,n}^\delta = \mu_{k,n}$ otherwise. We observe that $\lambda^\delta$ belongs to $B(\mu)$ if

$$\sum_{n\in[M]:k\neq k_n^\star} w_{n,m}\delta_n > \min_{k'\in[K]:k'\neq k_m^\star} \Delta_{k',m}' = \Delta_{k_m^\star,m}' \ ,$$

as this leads to $\sum_{n\in[M]} w_{n,m}\lambda_{k_m^\star,n} < \max_{k'\neq k_m^\star} \sum_n w_{n,m}\mu_{k',n}$ and arm $k_m^\star$ being sub-optimal in $\lambda^\delta$. For any $c \in \mathcal{C}$,

$$\min_{\delta:\sum_{n:k\neq k_n^\star} w_{n,m}\delta_n \geq \Delta_{k_m^\star,m}'} \sum_{n:k_n^\star\neq k_m^\star} c_{k,n}\frac{\delta_n^2}{2} \geq 1 \ ,$$

and Lemma 7 leads to

$$\sum_{n\in[M]:k_n^\star\neq k_m^\star} \frac{w_{n,m}^2}{c_{k,n}} \leq \frac{(\Delta_{k_m^\star,m}')^2}{2} \ .$$

In conclusion, for $c \in \mathcal{C}$, we proved that, for every $(k,m) \in [K] \times [M]$,

$$\sum_{n\in[M]:k_n^\star\neq k} \frac{w_{n,m}^2}{c_{k,n}} \leq \frac{(\Delta_{k,m}')^2}{2} \ ,$$

which proves the inclusion (5) and concludes the proof, as the minimum over a larger set is (potentially) smaller.

**Lemma 7.** *Let $\mathcal{N}$ be a set of indices, and $(c_n)_{n\in\mathcal{N}}$ be all positive. The minimizer over $\delta \in \mathbb{R}^{|\mathcal{N}|}$ of $\sum_{n\in\mathcal{N}} c_n \frac{\delta_n^2}{2}$, under the constraint that $\sum_{n\in\mathcal{N}} \delta_n w_{n,m} \geq d_m$, satisfies*

$$\forall n \in \mathcal{N}, \ \delta_n = \frac{d_m w_{n,m}/c_n}{\sum_{n'\in\mathcal{N}} \frac{w_{n',m}^2}{c_{n'}}} \ ,$$

*and the minimum is equal to*

$$\frac{d_m^2}{2}\left(\sum_{n\in\mathcal{N}} \frac{w_{n,m}^2}{c_n}\right)^{-1} \ .$$

Lemma 7 is proven using classical techniques for solving constrained minimization problems.

## B  Proof of Theorem 2

We recall the good event $\mathcal{E}$ defined in Lemma 3. First, the following lemma ensures the correctness of Algorithm 1 on $\mathcal{E}$, which holds with probability $1 - \delta$ from Lemma 3.

**Lemma 8.** *On event $\mathcal{E}$, when stopping, W-CPE-BAI outputs $\hat{k}_m = k_m^\star$ for each agent $m$.*

*Proof.* On event $\mathcal{E}$, it is not possible that one agent $m$ eliminates arm $k_m^\star$ from its set $B_m(r+1)$ at any round $r$; otherwise, if $j_m(r) \in \arg\max_{j\in B_m(r)}\{\hat{\mu}_{j,m}'(r) - \Omega_{j,m}(r)\}$, $j_m(r) \neq k_m^\star$, then the elimination criterion and event $\mathcal{E}$ imply that $\mu_{k_m^\star,m}' < \mu_{j_m(r),m}'$, which is absurd. $\qquad\square$

Then we upper bound the exploration cost when $\mathcal{E}$ holds. We denote by $R$ the (random) number of rounds used by the algorithm, and, for all $m \in [M]$ and $k \neq k_m^\star$, by $R_{k,m}$ the (random) last round in which $k$ is still a candidate arm for player $m$

$$R_{k,m} := \sup\{r \geq 0 : k \in B_m(r)\} \ .$$

By definition of Algorithm 1, $R = \max_m \max_{k \neq k_m^\star} R_{k,m}$. We first provide upper bounds on $R_{k,m}$ and $R$. To achieve this, we introduce the following notation for any $k \in [K], m \in [M]$,

$$r_{k,m} \ := \ \min \left\{ r \geq 0 : 4 \times 2^{-r} < \Delta'_{k,m} \right\} \ \text{and} \ r_{\max} := \max_{m \in [M]} \max_{k \neq k_m^\star} r_{k,m} \ .$$

The following upper bounds can be easily checked.

**Lemma 9.** $\forall k, m, r_{k,m} \leq \log_2 \left( 8/\Delta'_{k,m} \right)$ *and* $r_{\max} \leq \log_2 \left( 8/\Delta'_{\min} \right)$ .

Using the fact that W-CPE-BAI is only halving the proxy gaps of arms that are not eliminated, we can write down the value of the proxy gaps for these arms.

**Lemma 10.** $\forall m \in [M]$ *and* $k \in B_m(r)$, $\widetilde{\Delta}_{k,m}(r) = 2^{-r}$ .

Using the important relationship between proxy gaps and the confidence width as established in Lemma 4, we can further show that

**Lemma 11.** *On* $\mathcal{E}$, *for any* $m \in [M], k \neq k_m^\star$, $R_{k,m} \leq r_{k,m}$ .

*Proof.* Assume $\mathcal{E}$ holds. For any suboptimal arm $k$ for agent $m$, at round $r = r_{k,m}$, if $k \notin B_m(r)$, then $R_{k,m} < r_{k,m}$ . Otherwise, if $k \in B_m(r)$, then we know that $k_m^\star \in B_m(r)$, as event $\mathcal{E}$ holds (see the proof of Lemma 8). Then

$$
\begin{aligned}
\hat{\mu}'_{k,m}(r) + \Omega_{k,m}(r) &\leq_{(1)} \mu'_{k,m} + 2\Omega_{k,m}(r) \\
&\leq_{(2)} \mu'_{k,m} + 2\widetilde{\Delta}_{k,m}(r) = \mu'_{k,m} + 4\widetilde{\Delta}_{k,m}(r) - 2\widetilde{\Delta}_{k,m}(r) \\
&<_{(3)} \mu'_{k,m} + \Delta'_{k,m} - 2\widetilde{\Delta}_{k,m}(r) = \mu'_{k_m^\star,m} - 2\widetilde{\Delta}_{k,m}(r) \\
&\leq_{(1)} \hat{\mu}'_{k_m^\star,m}(r) + \Omega_{k_m^\star,m}(r) - 2\widetilde{\Delta}_{k,m}(r) \\
&\leq_{(2),(4)} \max_{j \in B_m(r)} \left\{ \hat{\mu}'_{j,m}(r) - \Omega_{j,m}(r) \right\} + 2 \cdot 2^{-r} - 2 \cdot 2^{-r} \ , \\
\implies \hat{\mu}'_{k,m}(r) + \Omega_{k,m}(r) &< \max_{j \in B_m(r)} \left\{ \hat{\mu}'_{j,m}(r) - \Omega_{j,m}(r) \right\} \ ,
\end{aligned}
$$

where (1) is using that event $\mathcal{E}$ holds ; (2) is using Lemma 4 ; (3) is using the definition of $r = r_{k,m}$, the fact that $k \in B_m(r)$ and Lemma 10 and (4) is using that $k, k_m^\star \in B_m(r)$ and Lemma 10. It follows that $k \notin B_m(r_{k,m} + 1)$ and $R_{k,m} \leq r_{k,m}$. $\qquad \square$

The previous lemma straightforwardly implies that

**Corollary 1.** $R \leq r_{\max} \leq \log_2(8/\Delta'_{\min})$ .

Moreover, it also permits to prove that, in the last round $R$, the proxy gaps are lower bounded by the gaps.

**Corollary 2.** *At final round $R$, and for any agent $m$ and suboptimal (with respect to $m$) arm $k \neq k_m^\star$, if $\mathcal{E}$ holds,*

$$\widetilde{\Delta}_{k,m}(R) \geq \frac{1}{8}\Delta'_{k,m} \ .$$

*Proof.* If $R < r_{k,m}$ , by definition of $r_{k,m}$ , we have that $\widetilde{\Delta}_{k,m}(R) \geq (1/4)\Delta'_{k,m} \geq (1/8)\Delta'_{k,m}$ . If $R \geq r_{k,m}$ , we first observe that $\widetilde{\Delta}_{k,m}(R) = \widetilde{\Delta}_{k,m}(R_{k,m}) = (1/2)\widetilde{\Delta}_{k,m}(R_{k,m} - 1)$ by definition of the algorithm (the gaps remain frozen when an arm is eliminated, and they are halved otherwise). As $R_{k,m} - 1 < r_{k,m}$ by Lemma 11, by definition of $r_{k,m}$ , it follows that

$$4\widetilde{\Delta}_{k,m}(R_{k,m} - 1) > \Delta'_{k,m}$$

and we conclude that $\widetilde{\Delta}_{k,m}(R) \geq (1/8)\Delta'_{k,m}$ . $\qquad \square$

Note that, for any $m \in [M]$, $\widetilde{\Delta}_{k_m^\star, m}(R) = \min_{k \ne k_m^\star} \widetilde{\Delta}_{k,m}(R)$ by Algorithm 1. Now, for any $m \in [M], k \in [K]$, using Corollary 2, and the fact that the proxy gaps are non-increasing between two consecutive phases, we get

$$\forall k \in [K], \forall m \in [M], \forall r \le R, \ \widetilde{\Delta}_{k,m}(r) \ge \widetilde{\Delta}_{k,m}(R) \ge \frac{\Delta'_{k,m}}{8} \ .$$

Using Lemma 2 and the fact that proxy gaps are non-increasing, for any round $r \le R$, the optimal allocation $t(r) \in \widetilde{\mathcal{P}}^\star \left( \left( \sqrt{2} \widetilde{\Delta}_{k,m}(r) \right)_{k,m} \right)$ satisfies

$$\sum_{k,m} t_{k,m}(r) \le 32 \sum_{k,m} t'_{k,m} \ ,$$

where $t' \in \widetilde{\mathcal{P}}^\star(\Delta')$ , hence

$$\max_{r \le R} \left[ \sum_{k,m} t_{k,m}(r) \right] \le 32 \widetilde{T}_W^\star(\mu) \ . \tag{6}$$

For every $k \in [K], m \in [M]$, we now introduce

$$r'_{k,m} := \sup\{r \le R : d_{k,m}(r) \ne 0\} \ ,$$

so that $n_{k,m}(R) = n_{k,m}(r'_{k,m})$ . Using Lemma 12 stated below, and the fact that function $\beta_\delta$ is nondecreasing in each coefficient of its argument (see its definition in Lemma 3),

$$
\begin{aligned}
n_{k,m}(R) = n_{k,m}(r'_{k,m}) &\le t_{k,m}(r'_{k,m}) \beta_\delta(n_{k,\cdot}(r'_{k,m})) + 1 \\
&\le t_{k,m}(r'_{k,m}) \beta_\delta(n_{k,\cdot}(R)) + 1 \ .
\end{aligned}
$$

**Lemma 12.** *For any $k, m, r \ge 0$, either $d_{k,m}(r) = 0$, or $n_{k,m}(r) = n_{k,m}(r-1) + d_{k,m}(r) < t_{k,m}(r) \beta_\delta(n_{k,\cdot}(r)) + 1$ .*

*Proof.* At fixed $r \ge 0$, for any set $S \subseteq [K] \times [M]$, let us prove by induction on $|S| \ge 1$ [7]

$$
\begin{aligned}
\forall k, m, \quad & d'_{k,m}(r) := (d_{k,m}(r) - \mathbb{1}_S((k,m)))_+ \\
\implies \quad & \forall (k,m) \in S, \ \frac{n_{k,m}(r-1) + d'_{k,m}(r)}{\beta_\delta(n_{k,\cdot}(r-1) + d_{k,\cdot}(r))} < t_{k,m}(r) \\
& \text{or } d_{k,m}(r) = 0 \ .
\end{aligned}
$$

**At $|S| = 1$** : Let us denote $S = \{(k', m')\}$ . If $d_{k',m'}(r) = 0$ , then it is trivial. Otherwise, $\sum_{k,m} d'_{k,m}(r) < \sum_{k,m} d_{k,m}(r)$ , and then, by minimality of solution $d(r)$ , at least one constraint from the optimization problem of value $\sum_{k,m} d_{k,m}(r)$ has to be violated. For any $(k,m) \notin S$ , by definition of $d(r)$ and nondecreasingness of $\beta_\delta$ ,

$$
\begin{aligned}
n_{k,m}(r-1) + d'_{k,m}(r) &= n_{k,m}(r-1) + d_{k,m}(r) \\
&\ge t_{k,m}(r) \beta_\delta(n_{k,\cdot}(r-1) + d_{k,\cdot}(r)) \\
&\ge t_{k,m}(r) \beta_\delta(n_{k,\cdot}(r-1) + d'_{k,\cdot}(r)) \ .
\end{aligned}
$$

That means, necessarily the only constraint that is violated is the one on $(k', m')$ . Using the nondecreasingness of $\beta_\delta$ :

$$
\begin{aligned}
n_{k',m'}(r-1) + d_{k',m'}(r) - 1 &= n_{k',m'}(r-1) + d'_{k',m'}(r) \\
&< t_{k',m'}(r) \beta_\delta(n_{k',\cdot}(r-1) + d'_{k,\cdot}(r)) \\
&\le t_{k',m'}(r) \beta_\delta(n_{k',\cdot}(r-1) + d_{k,\cdot}(r)) \ .
\end{aligned}
$$

Combining the two ends of the inequality proves the claim.

---

[7] For any $x \in \mathbb{N}^M$ , $(x)_+ := (\max(0, x_m))_{m \in [M]}$ , and $\mathbb{1}_S$ is the indicator function of set $S$ .

**At $|S| > 1$ :** At fixed $(k', m') \in S$ , we can apply the claim to $S \smallsetminus \{(k', m')\}$. Moreover, if $d_{k',m'}(r) = 0$ , then the claim is proven. Otherwise, by appealing to the extremes,

$$n_{k',m'}(r-1) + d'_{k',m'}(r) \ge t_{k',m'}(r)\beta_\delta(n_{k',\cdot}(r-1) + d_{k,\cdot}(r)) .$$

Let us then consider the following allocation:

$$\forall k, m, d''_{k,m}(r) := d_{k,m}(r) - \mathbb{1}_{\{(k',m')\}}((k,m)) .$$

It can be checked straightforwardly – using the nondecreasingness of $\beta_\delta$ – that $d''$ satisfies all required constraints for any pair $(k, m) \in [K] \times [M]$ , and that $\sum_{k,m} d''_{k,m}(r) = \sum_{k,m} d_{k,m}(r) - 1$ , which, by minimality of $d$ , is absurd. Then the claim is proven for $|S| > 1$ . Then Lemma 12 is proven by considering $S = [K] \times [M]$ .

$\square$

By summing the upper bound on $(n_{k,m}(R))_{k,m}$ over $[K] \times [M]$, we can upper bound the exploration cost $\tau$ as

$$
\begin{aligned}
\tau := \sum_{k,m} n_{k,m}(R) \quad &\le \quad \sum_{k,m} t_{k,m}(r'_{k,m})\beta_\delta(n_{k,\cdot}(R)) + KM \\
&\le \quad \sum_{k,m} t_{k,m}(r'_{k,m})\beta^*(\tau) + KM \\
&\le \quad \sum_{k,m} \sum_{r \le R} t_{k,m}(r)\beta^*(\tau) + KM \\
&\le \quad R \max_{r \le R} \left[ \sum_{k,m} t_{k,m}(r) \right] \beta^*(\tau) + KM \\
&\le \quad \log_2(8/\Delta'_{\min}) \max_{r \le R} \left[ \sum_{k,m} t_{k,m}(r) \right] \beta^*(\tau) + KM
\end{aligned}
$$

where we use Corollary 1 and introduce the quantity

$$\beta^*(\tau) := \beta_\delta(\tau \mathbb{1}_M) = 2\left(g_M\left(\frac{\delta}{KM}\right) + 2M \ln(4 + \ln(\tau))\right) \text{ where } \forall n \in [M], \mathbb{1}_M(n) = 1 .$$

Using Equation (6) and Lemma 1,

$$\tau \le 32\widetilde{T}^\star_W(\mu) \log_2(8/\Delta'_{\min})\beta^*(\tau) + KM \le 32T^\star_W(\mu) \log_2(8/\Delta'_{\min})\beta^*(\mu) + KM .$$

Therefore, $\tau$ is upper bounded by

$$\sup\{n \in \mathbb{N}^\star : n \le 32T^\star_W(\mu) \log_2(8/\Delta'_{\min})\beta^*(n) + KM\} .$$

Applying Lemma 15 in [24] with

$$
\begin{aligned}
\Delta &= \left(\sqrt{32T^\star_W(\mu) \log_2(8/\Delta'_{\min})}\right)^{-1} , \\
a &= KM + 2g_M\left(\frac{\delta}{KM}\right) , \\
b &= 4M , \\
c &= 4 , \\
d &= e^{-1} \text{ ( using } \forall n, \log(n) \le ne^{-1} \text{ )} ,
\end{aligned}
$$

$\tau$ is upper bounded by

$$
\begin{aligned}
\hat{T}_W(\mu) \quad := \quad &32T^\star_W(\mu) \log_2(8/\Delta'_{\min})\left[ KM + 2g_M\left(\frac{\delta}{KM}\right) \right. \\
&\left. + 4M \ln\left(4 + 1,024\frac{(T^\star_W(\mu) \log_2(8/\Delta'_{\min}))^2}{e}\left(KM + 2g_M\left(\frac{\delta}{KM}\right) + 4M(2 + \sqrt{e})\right)^2\right)\right] ,
\end{aligned}
$$

which satisfies $\tau \hat{T}_W(\mu) \leq a + b \ln(c + d\tau \hat{T}_W(\mu))$. Using that $g_M(x) \simeq x + M \log\log(x)$ in the regime of small values of $\delta$, we obtain that

$$\hat{T}_W(\mu) = 32 T_W^\star(\mu) \log_2\left(8/\Delta_{\min}'\right) \log(1/\delta) + o_{\delta \to 0}\left(\log(1/\delta)\right).$$

The upper bound on the communication cost follows from the upper bound on the number of phases given in Corollary 1.

## C   Supplementary Lemmas and Proofs

We report here technical lemmas and their proofs, ordered by section.

### C.1   Collaborative Best-Arm Identification

**Lemma 13.** *Introducing the quantity*

$$\widetilde{T}_W^\star(\mu) := \min_{t \in (\mathbb{R}^+)^{K \times M}} \sum_{(k,m) \in [K] \times [M]} t_{k,m} \ \text{s.t.} \ \forall m \in [M], k \in [K], \ \sum_{n \in [M]} \frac{w_{n,m}^2}{t_{k,n}} \leq \frac{\left(\Delta_{k,m}'\right)^2}{2},$$

*it holds that $\widetilde{T}_W^\star(\mu) \leq T_W^\star(\mu) \leq 2\widetilde{T}_W^\star(\mu)$.*

*Proof.* Let us denote by $\mathcal{C}$ and $\widetilde{\mathcal{C}}$ the two constraint sets such that $T_W^\star(\mu) = \min\left\{\sum_{k,m} t_{k,m} \mid t \in \mathcal{C}\right\}$ and $\widetilde{T}_W^\star(\mu) = \min\left\{\sum_{k,m} t_{k,m} \mid t \in \widetilde{\mathcal{C}}\right\}$. The inequality $\widetilde{T}_W^\star(\mu) \leq T_W^\star(\mu)$ is obtained by noticing that $\mathcal{C} \subseteq \widetilde{\mathcal{C}}$. To prove the other inequality, we consider $\widetilde{\tau} \in \arg\min\left\{\sum_{k,m} t_{k,m} \mid t \in \widetilde{\mathcal{C}}\right\}$. Then, for any agent $m \in [M]$, arm $k \neq k_m^\star$,

$$\sum_{n \in [M]} w_{n,m}^2 \left(\frac{1}{2\widetilde{\tau}_{k,n}} + \frac{1}{2\widetilde{\tau}_{k_m^\star,n}}\right) = \frac{1}{2}\underbrace{\left(\sum_{n \in [M]} \frac{w_{n,m}^2}{\widetilde{\tau}_{k,n}}\right)}_{\leq \left(\Delta_{k,m}'\right)^2/2} + \frac{1}{2}\underbrace{\left(\sum_{n \in [M]} \frac{w_{n,m}^2}{\widetilde{\tau}_{k_m^\star,n}}\right)}_{\leq \left(\Delta_{k_m^\star,m}'\right)^2/2 := \min\left\{\left(\Delta_{k',m}'\right)^2/2 \mid k' \neq k_m^\star\right\}}$$

$$\leq \left(\Delta_{k,m}'\right)^2/2.$$

Then $2\widetilde{\tau} \in \mathcal{C}$, therefore by minimality, $T_W^\star(\mu) \leq 2\widetilde{T}_W^\star(\mu)$. □

**Lemma 14.** *Consider $\Delta, \Delta' \in (\mathbb{R}^+)^{K \times M}$, such that $\tau \in \widetilde{\mathcal{P}}^\star(\Delta)$ and $\tau' \in \widetilde{\mathcal{P}}^\star(\Delta')$. Then*

**(i).** *If there exists $\alpha > 0$ such that: $\forall k \in [K], \forall m \in [M], \alpha\Delta_{k,m} \leq \Delta_{k,m}'$,*

$$\sum_{k,m} \tau_{k,m}' \leq \frac{1}{\alpha^2} \sum_{k,m} \tau_{k,m}.$$

**(ii).** *If there exists $\beta > 0$ such that: $\forall k \in [K], \forall m \in [M], \Delta_{k,m}' \leq \beta\Delta_{k,m}$. Then*

$$\frac{1}{\beta^2} \sum_{k,m} \tau_{k,m} \leq \sum_{k,m} \tau_{k,m}'.$$

*Proof.* The proof follows from the fact that $\tau$ and $\tau'$ are *minimal*. In particular, to prove **(ii)**, let $\tau_{k,m}'' = \beta^2 \tau_{k,m}'$ for any $k \in [K]$, $m \in [M]$. Then, for any agent $m$ and arm $k$,

$$\sum_{n \in [M]} \frac{w_{n,m}^2}{\tau_{k,n}''} = \sum_{n \in [M]} \frac{w_{n,m}^2}{\beta^2 \tau_{k,n}'} \leq \frac{1}{2}\left(\frac{\Delta_{k,m}'}{\beta}\right)^2 \leq \frac{\Delta_{k,m}^2}{2}.$$

By minimality of $\tau$,

$$\sum_{m \in [M]} \tau_{k,m} \leq \sum_{m \in [M]} \tau_{k,m}'' = \beta^2 \sum_{n \in [M]} \tau_{k,m}'.$$

**(i)** similarly follows. □

## C.2 A Near-Optimal Algorithm For Best Arm Identification

**Lemma 15.** *Let us define*

$$\beta_\delta(N) := 2\left(g_M\left(\frac{\delta}{KM}\right) + 2\sum_{m=1}^{M}\ln(4 + \ln(N_m))\right),$$

*for any $N \in (\mathbb{N}^*)^M$, where*

$$\forall \delta \in (0,1), g_M(\delta) := M\mathcal{C}^{g_G}\left(\log(1/\delta)/M\right),$$

$$\forall x > 0, \mathcal{C}^{g_G}(x) := \min_{\lambda \in (0.5,1)}\frac{g_G(\lambda) + x}{\lambda},$$

*and $\forall \lambda \in (0.5,1), g_G(\lambda) := 2\lambda - 2\lambda\log(4\lambda) + \log(\zeta(2\lambda)) - 0.5\log(1 - \lambda)$,*

*where $\zeta$ is the Riemann zeta function. Then, the good event*

$$\mathcal{E} := \left\{\forall r \in \mathbb{N}, \forall m, \forall k, \left|\hat{\mu}'_{k,m}(r) - \mu'_{k,m}\right| \le \Omega_{k,m}(r)\right\}.$$

*holds with probability larger than $1 - \delta$.*

*Proof.* Using Proposition 24 from [23] on $\mu'_{k,m}$, for any arm $k$ and agent $m$, directly yields

$$\mathbb{P}\left(\exists r \ge 0, |\hat{\mu}'_{k,m}(r) - \mu'_{k,m}| > \sqrt{2\left(g_M\left(\frac{\delta}{KM}\right) + 2\sum_{n=1}^{M}\ln(4 + \ln(n_{k,n}(r)))\right)\sum_{n}\frac{w^2_{n,m}}{n_{k,n}(r)}}\right) \le \frac{\delta}{KM}$$

(using the notation of the paper, consider $\mu = \mu_{k,\cdot}$ and $c = W_{\cdot,m}$). Then all that is needed to conclude is to apply a union bound on $[K] \times [M]$

$$\mathbb{P}(\mathcal{E}^c) = \mathbb{P}\left(\exists m \in [M], \exists k \in [K], \exists r \ge 0, |\hat{\mu}'_{k,m}(r) - \mu'_{k,m}| > \sqrt{2\beta_\delta(n_{k,\cdot}(r))\sum_{n}\frac{w^2_{n,m}}{n_{k,n}(r)}}\right)$$

$$\le \sum_{m\in[M]}\sum_{k\in[K]}\frac{\delta}{KM} \le \delta.$$

$\square$

## C.3 Regret Lower Bound

**Lemma 16.** *Introducing the quantity*

$$\widetilde{C}^\star_W(\mu) := \min_{c\in(\mathbb{R}^+)^{K\times M}}\sum_{k=1}^{K}\sum_{m=1}^{M}c_{k,m}\Delta'_{k,m} \text{ s.t. } \forall k \in [K], \forall m \in [M], \sum_{n=1}^{M}\frac{w^2_{n,m}}{c_{k,n}} \le \frac{\Delta'_{k,m}}{2\sigma^2},$$

*it holds that $C^\star_W(\mu) \le \widetilde{C}^\star_W(\mu) \le 4C^\star_W(\mu)$.*

*Proof.* Let $c$ and $\widetilde{c}$ be the solutions to the optimization problems of $C^\star_W(\mu)$ and $\widetilde{C}^\star_W(\mu)$, respectively. Note that, for any agent $m$, $c_{m,k^\star_m} = +\infty$ because, in the optimization problem related to the regret lower bound, these terms do not contribute to the objective. The lower bound follows from the definition of $c$ and the fact that $(\widetilde{c}_{k,m})_{m,k\ne k^\star_m}$ satisfy the same constraints as $(\widetilde{c}_{k,m})_{m,k\ne k^\star_m}$.

Next, we prove the upper bound on $\widetilde{C}^\star_W(\mu)$. For any $k \in [K]$, define $\mathcal{S}^*_k = \{m : k = k^\star_m\}$ and $\mathcal{S}_k = \{m : k \ne k^\star_m\}$ (note that for any $k \in [K]$, $\{\mathcal{S}^*_k, \mathcal{S}_k\}$ is a partition of $[M]$). For $k \in [K]$ and $m \in \mathcal{S}_k$, let $c'_{k,m} = 2c_{k,m}$. For $k \in [K]$ and $m \in \mathcal{S}^*_k$, let $c'_{k,m} = c^1_{k,m}$, where

$$c^1 \in \arg\min_{\tau\in(\mathbb{R}^+)^{K\times M}}\sum_{k=1}^{K}\sum_{m\in\mathcal{S}^*_k}\tau_{k,m}\Delta'_{k,m} \text{ s.t.} \forall k \in [K], m \in [M], \sum_{n\in\mathcal{S}^*_k}\frac{w^2_{n,m}}{\tau_{k,n}} \le \frac{(\Delta'_{k,m})^2}{4}.$$

Note that $c'_{k,m}$ satisfy the same constraints as $\widetilde{c}_{k,m}$: $\forall k \in [K], m \in [M], \sum_{n=1}^{M} \frac{w_{n,m}^2}{c'_{k,n}} \leq \frac{(\Delta'_{k,m})^2}{2}$ . We thus have

$$
\begin{aligned}
\sum_{k=1}^{K} \sum_{m=1}^{M} \widetilde{c}_{k,m} \Delta'_{k,m} \quad &\leq \quad \sum_{k=1}^{K} \sum_{m=1}^{M} c'_{k,m} \Delta_{k,m} \\
&= \quad \sum_{k=1}^{K} \sum_{m \in \mathcal{S}_k} c'_{k,m} \Delta_{k,m} + \sum_{k=1}^{K} \sum_{m \in \mathcal{S}_k^*} c'_{k,m} \Delta_{k,m} \\
&= \quad 2 \sum_{k=1}^{K} \sum_{m \in \mathcal{S}_k} c_{k,m} \Delta'_{k,m} + \sum_{k=1}^{K} \sum_{m \in \mathcal{S}_k^*} c'_{k,m} \Delta_{k,m} \\
&= \quad 2 C_W^\star(\mu) + \sum_{k=1}^{K} \sum_{m \in \mathcal{S}_k^*} c_{k,m}^1 \Delta'_{k,m} \\
&\leq \quad 4 C_W^\star(\mu) \ .
\end{aligned}
$$

The first inequality holds by minimality of $\widetilde{c}$. To prove the second inequality, for all $m \in [M]$, let us define $c''_{k_m^\star,m} = 2 \sum_{k \neq k_m^\star} c_{k,m}$ , and $c''_{k,m} = \infty$ for $k \neq k_m^\star$ . Note that $c''$ satisfy the constraints in the definition of $c^1$ . Thus, by minimality of $c^1$ , and using the fact that, for any $m \in [M]$ , $\Delta'_{k_m^\star,m} := \min_{k \neq k_m^\star} \Delta'_{k,m}$ , we have

$$
\begin{aligned}
\sum_{k=1}^{K} \sum_{m \in \mathcal{S}_k^*} c_{k,m}^1 \Delta'_{k,m} \quad &\leq \quad \sum_{k=1}^{K} \sum_{m \in \mathcal{S}_k^*} c''_{k,m} \Delta'_{k,m} \\
&= \quad 2 \sum_{k=1}^{K} \sum_{k,m \in \mathcal{S}_k^*} \left( \sum_{j \neq k_m^\star} c_{j,m} \right) \Delta'_{k,m} \\
&= \quad 2 \sum_{k=1}^{K} \sum_{m \in \mathcal{S}_k} c_{k,m} \Delta'_{k_m^\star,m} \\
&\leq \quad 2 \sum_{k=1}^{K} \sum_{m \in \mathcal{S}_k} c_{k,m} \Delta'_{k,m} \\
&= \quad 2 C_W^\star(\mu) \ ,
\end{aligned}
$$

which completes the proof.

$\square$

# D   Extension to Top-$N$ Identification

A generalization of the best arm identification problem is Top-$N$ identification, which is the problem of finding the $N$ optimal arms (for mixed rewards) for each agent. For any model $\mu \in \mathbb{R}^{K \times M}$, weight matrix $W \in [0,1]^{M \times M}$ , such that $\mu' = \mu W$ , any agent $m$ , and positive integer $N \leq K$ , let us define [8]

$$
S_m^\star := \left\{ k \in [K] \mid \mu'_{k,m} \geq \max_{k' \in [K]}^N \mu'_{k',m} = \max_{k' \in [K]}^N \sum_{n \in [M]} w_{n,m} \mu_{k',n} \right\} \ .
$$

In this case, an algorithm for Top-$N$ identification returns at the end of the exploration phase a set of $N$ arms denoted $\hat{S}_m$ for agent $m$ . $\delta$-correctness is defined as follows

---

[8]We define operation $\max^N$ such that $\max_{i \in S}^N f(i)$ is the $N^{th}$ (without multiplicity) greatest value in set $\{f(i) : i \in S\}$ for any function $f : S \mapsto \mathbb{R}$.

$$\mathbb{P}_\mu \left( \forall m \in [M], \hat{S}_m \subseteq S_m^\star \right) \geq 1 - \delta \,.$$

Note that there might be more than $N$ arms in a given set $S_m^\star, m \in [M]$. Similarly to best arm identification, we also assume here that the set of top-$N$ arms is unique – that is, for any $m, |S_m^\star| = N$.

**Lower Bound for Top-$N$ Identification** For Top-$N$ identification, one can prove, similarly to the proof of Theorem 1 – using Lemma 1 in [28] to define the set of alternative models – the following result, which is valid for Gaussian rewards with fixed variance $\sigma^2 = 1$, and any weight matrix $W$ such that all diagonal coefficients are positive,

**Theorem 4.** *Let $\mu$ be a fixed matrix of means in $\mathbb{R}^{K \times M}$. For any $\delta \in (0, 1/2]$, let $\mathcal{A}$ be a $\delta$-correct algorithm under which each agent communicates each reward to the central server after it is observed, and let us denote for any $k \in [K], m \in [M]$, $\tau_{k,m} := \mathbb{E}_\mu^{\mathcal{A}}[N_{k,m}(\tau)]$, where $\tau$ is the stopping time. For any $m \in [M], k \notin S_m^\star, l \in S_m^\star$, denoting $\mu' = \mu W$, it holds that*

$$\sum_n w_{n,m}^2 \left( \frac{1}{\tau_{k,n}} + \frac{1}{\tau_{l,n}} \right) \leq \frac{\left( \mu'_{k,m} - \mu'_{l,m} \right)^2}{2 \log(1/(2.4\delta))} \,,$$

*and therefore $Exp_\mu(\mathcal{A}) \geq N_W^\star(\mu) \log \left( \frac{1}{2.4\delta} \right)$, where*

$$N_W^\star(\mu) := \min_{t \in \mathbb{R}^{K \times M}} \sum_{(k,m) \in [K] \times [M]} t_{k,m}$$

$$s.t. \ \forall m, k \notin S_m^\star(\mu), l \in S_m^\star(\mu), \sum_{n \in [M]} w_{n,m}^2 \left( \frac{1}{t_{k,n}} + \frac{1}{t_{l,n}} \right) \leq \frac{\left( \mu'_{k,m} - \mu'_{l,m} \right)^2}{2} \,.$$

*Proof.* As mentioned, let us use Lemma 1 from [28] to define the set of alternative models to $\mu$ in Top-$N$ identification. If, for any agent $m$, $S_m^\star(\mu)$ is the set of its top-$N$ arms (of size $N$) with respect to mixed rewards, then

$$\begin{aligned} \text{Alt}(\mu) &:= \left\{ \lambda \in \mathbb{R}^{K \times M} : \exists m, S_m^\star(\mu) \nsubseteq S_m^\star(\lambda) \right\} \\ &= \left\{ \lambda \in \mathbb{R}^{K \times M} : \exists m, \exists k \notin S_m^\star(\mu), \exists l \in S_m^\star(\mu) : \lambda'_{k,m} > \lambda'_{l,m} \right\} \,, \end{aligned}$$

where $\lambda'_{k,m} := \sum_{n \in [M]} w_{n,m} \lambda_{k,n}$ for any arm $k$ and agent $m$. If we assume that stopping time $\tau$ is almost surely finite under $\mu$ for algorithm $\mathcal{A}$, then let event $\mathcal{E}_\mu := \left\{ \exists m : \hat{S}_m \nsubseteq S_m^\star(\mu) \right\}$. Using the $\delta$-correctness of algorithm $\mathcal{A}$, where $\delta \leq 1/2$, by Theorem 1 from [15],

$$\frac{1}{2} \sum_{k,m} \tau_{k,m} (\mu_{k,m} - \lambda_{k,m})^2 \geq \log \left( \frac{1}{2.4\delta} \right) \,. \tag{7}$$

Similarly to the best arm identification case, we can show that, since all diagonal coefficients of $W$ are positive, for any $k \in [K], m \in [M]$, $\tau_{k,m} > 0$. Consider now a fixed agent $m$, and two arms $k \notin S_m^\star(\mu), l \in S_m^\star(\mu)$. We will build an alternative model $\lambda$, similar enough to $\mu$, where only arms $k$ and $l$ are modified for all agents, such that $l \notin S_m^\star(\lambda)$ and $k \in S_m^\star(\lambda)$. The procedure is similar to what we did in best arm identification. Given two nonnegative sequences $(\delta_n)_{n \in [M]}$ and $(\delta'_n)_{n \in [M]}$, we define $\lambda = (\lambda_{k',n})_{k' \in [K]}$ such that

$$\begin{cases} \lambda_{k',n} &= \mu_{k',n} \text{ if } k' \notin \{k, l\} \,, \\ \lambda_{k,n} &= \mu_{k,n} + \delta_n \,, \\ \lambda_{l,n} &= \mu_{l,n} - \delta'_n \,, \end{cases}$$

and which satisfies

$$(\lambda'_{k,m} - \mu'_{k,m}) - (\lambda'_{l,m} - \mu'_{l,m}) = \sum_{n \in [M]} w_{n,m} (\delta_n + \delta'_n) \geq \mu'_{l,m} - \mu'_{k,m} \,. \tag{8}$$

From Equation (7),

$$\inf_{\delta, \delta' : (8) \text{ holds}} \left[ \sum_n \tau_{k,n} \frac{\delta_n^2}{2} + \sum_n \tau_{l,n} \frac{(\delta'_n)^2}{2} \right] \,.$$

Solving this constrained optimization problem yields the following solution

$$\delta_n = \frac{(\mu'_{l,m} - \mu'_{k,m})w_{n,m}/\tau_{k,n}}{\sum_{n' \in [M]} w^2_{n',m} (1/\tau_{k,n'} + 1/\tau_{l,n'})} \text{ and } \delta'_n = \frac{(\mu'_{l,m} - \mu'_{k,m})w_{n,m}/\tau_{l,n}}{\sum_{n' \in [M]} w^2_{n',m} (1/\tau_{k,n'} + 1/\tau_{l,n'})} .$$

We conclude similarly to the best arm identification case. □

Note that we retrieve the same bound as for the case $N = 1$ (*i.e.*, best arm identification).

**Relaxed Lower Bound Problem for Top-$N$ Identification**   For Top-$N$ identification, let us define

$$\forall k, m, \Delta'^N_{k,m} := \begin{cases} \max^N_{k' \in [K]} \mu'_{k',m} - \mu'_{k,m} & \text{if } k \notin S^\star_m \\ \mu'_{k,m} - \max^{N+1}_{k' \in [K]} \mu'_{k',m} & \text{otherwise} , \end{cases}$$

and $\widetilde{N}^\star_W(\mu)$ the value of problem $\widetilde{\mathcal{P}}^\star \left( \left( (\Delta'^N_{k,m})^2/2 \right)_{k,m} \right)$. The set of constraints $\mathcal{N} := \left\{ t \mid \forall m \in [M], \forall k \notin S^\star_m, \forall l \in S^\star_m, \sum_n w^2_{n,m} \left( \frac{1}{t_{k,n}} + \frac{1}{t_{l,n}} \right) \leq \frac{(\mu'_{k,m} - \mu'_{l,m})^2}{2} \right\}$ is included in the set of constraints $\widetilde{\mathcal{N}} := \left\{ t \mid \forall m \in [M], \forall k \in [K], \sum_n \frac{w^2_{n,m}}{t_{k,n}} \leq \frac{(\Delta'^N_{k,m})^2}{2} \right\}$ : indeed, if $t \in \mathcal{N}$, then for any $m \in [M]$, and any $k \notin S^\star_m$,

$$\forall l \in S^\star_m, \ \sum_n \frac{w^2_{n,m}}{t_{k,n}} \leq \sum_n w^2_{n,m} \left( \frac{1}{t_{k,n}} + \frac{1}{t_{l,n}} \right) \leq \frac{(\mu'_{k,m} - \mu'_{l,m})^2}{2}$$

$$\implies \sum_n \frac{w^2_{n,m}}{t_{k,n}} \leq \min_{l \in S^\star_m} \frac{(\mu'_{k,m} - \mu'_{l,m})^2}{2} = \frac{(\mu'_{k,m} - \max^N_{k' \in [K]} \mu'_{k',m})^2}{2} = \frac{(\Delta'^N_{k,m})^2}{2} .$$

Similarly, for any agent $m$ and $l \in S^\star_m(\mu)$, one can check that $\sum_n \frac{w^2_{n,m}}{t_{l,n}} \leq \frac{(\Delta'^N_{l,m})^2}{2}$, hence $t \in \widetilde{\mathcal{N}}$. Then $N^\star_W(\mu) \geq \widetilde{N}^\star_W(\mu)$.

**Algorithm for Top-$N$ identification**   Algorithm 1 can then easily be adapted to Top-$N$ identification, with the following changes (the full algorithm is described in Algorithm 2)

**1.** Replace the stopping criterion (and the condition for the update of proxy gaps $(\widetilde{\Delta}_{k,m}(r))_{k,m}$ at round $r$) with
$$\forall m \in [M], |B_m(r)| \leq N ,$$

**2.** Replace the elimination criterion with

$$B_m(r+1) \leftarrow \left\{ k \in B_m(r) \mid \mu'_{k,m} + \Omega_{k,m}(r) \geq \max^N_{i \in B_m(r)} \left( \mu'_{i,m} - \Omega_{i,m}(r) \right) \right\} .$$

**Remark 2.** *Note that the proxy gap $\widetilde{\Delta}_{k,m}(r)$ no longer tracks the value of gap $\Delta'_{k,m}$ for $m \in [M], k \in [K]$, but $\Delta'^N_{k,m}$, and that on $N = 1$, this algorithm exactly coincides with Algorithm 1.*

**Analysis of Algorithm 2.**   First, such an algorithm is indeed $\delta$-correct on event $\mathcal{E}$ (the same as defined for Algorithm 1). Otherwise, for some agent $m$, there would be an arm $l \in S^\star_m$ which is eliminated at round $r$ from $B_m(r+1)$. But, on event $\mathcal{E}$, Lemma 3 implies that, for any $r \geq 0$, $m \in [M]$, and $(i,j) \in [K]^2$,

$$\hat{\mu}'_{i,m}(r) - \hat{\mu}'_{j,m}(r) + \Omega_{i,m}(r) + \Omega_{j,m}(r) \geq \mu'_{i,m} - \mu'_{j,m} \geq \hat{\mu}'_{i,m}(r) - \hat{\mu}'_{j,m}(r) - \Omega_{i,m}(r) - \Omega_{j,m}(r) .$$

**Algorithm 2** Weighted Collaborative Phased Elimination for Top-$N$ identification (W-CPE-Top$N$)

---

**Input:** $\delta \in (0, 1)$, $M$ agents, $K$ arms, matrix $W$, $N \in [K]$
Initialize $r \leftarrow 0$, $\forall k, m, \widetilde{\Delta}_{k,m}(0) \leftarrow 1, n_{k,m}(0) \leftarrow 1, \forall m, B_m(0) \leftarrow [K]$
Draw each arm $k$ by each agent $m$ once
**repeat**
  # Central server
  $B(r) \leftarrow \bigcup_{m \in [M]} B_m(r)$
  Compute $t(r) \leftarrow \widetilde{\mathcal{P}}^\star \left( \left( \sqrt{2} \widetilde{\Delta}_{k,m}(r) \right)_{k,m} \right)$
  For all $k \in [K]$, compute

$$(d_{k,m}(r))_{m \in [M]} \leftarrow \arg\min_{d \in \mathbb{N}^M} \sum_m d_m \text{ s.t. } \forall m \in [M], \frac{n_{k,m}(r-1) + d_m}{\beta_\delta(n_{k,\cdot}(r-1) + d)} \geq t_{k,m}(r)$$

  Send to each agent $m$ $(d_{k,m}(r))_{k,m}$ and $d_{\max} := \max_{n \in [M]} \sum_{k \in [K]} d_{k,n}(r)$

  # Agent $m$
  Sample arm $k \in B(r)$ $d_{k,m}(r)$ times, so that $n_{k,m}(r) = n_{k,m}(r-1) + d_{k,m}(r)$
  Remain idle for $d_{\max} - \sum_{k \in [K]} d_{k,m}(r)$ rounds
  Send to the server empirical mean $\hat{\mu}_{k,m}(r) := \sum_{s \leq n_{k,m}(r)} X_{k,m}(s)/n_{k,m}(r)$ for any $k \in [K]$

  # Central server
  Compute the empirical mixed means $(\hat{\mu}'_{k,m}(r))_{k,m}$ based on $(\hat{\mu}_{k,m}(r))_{k,m}$ and $W$
  *// Update set of candidate best arms for each user*
  **for** $m = 1$ **to** $M$ **do**

$$B_m(r+1) \leftarrow \left\{ k \in B_m(r) \mid \hat{\mu}'_{k,m}(r) + \Omega_{k,m}(r) \geq \max_{j \in B_m(r)}^{N} \left( \hat{\mu}'_{j,m}(r) - \Omega_{j,m}(r) \right) \right\}$$

  **end for**
  *// Update the gap estimates*
  For all $k, m$, $\widetilde{\Delta}_{k,m}(r+1) \leftarrow \widetilde{\Delta}_{k,m}(r) \times (1/2)^{\mathbb{1}(k \in B_m(r+1) \wedge |B_m(r+1)| > N)}$
  $r \leftarrow r + 1$
**until** $\forall m \in [M], |B_m(r)| \leq N$
**Output:** $\{k \in B_m(r) : m \in [M]\}$

---

Then, combining the right-hand inequality for $j = l$ with the elimination criterion

$$\begin{aligned}
\max_{i \in [K]}^{N} \mu'_{i,m} - \mu'_{l,m} &\geq \max_{i \in [K]}^{N} (\hat{\mu}'_{i,m}(r) - \hat{\mu}'_{l,m}(r) - \Omega_{i,m}(r) - \Omega_{l,m}(r)) \\
&\geq \max_{i \in B_m(r) \subseteq [K]}^{N} (\hat{\mu}'_{i,m}(r) - \hat{\mu}'_{l,m}(r) - \Omega_{i,m}(r) - \Omega_{l,m}(r)) > 0 ,
\end{aligned}$$

which is absurd because $l \in S_m^\star$. Then, let us consider the following notation, for any $m \in [M]$, $k \notin S_m^\star$,

$$R_{k,m} := \sup\{r \geq 0 : k \in B_m(r)\} \text{ and } r_{k,m}^N := \min\left\{r \geq 0 : 4\widetilde{\Delta}_{k,m}(r) < \Delta_{k,m}^{'N}\right\} .$$

Note that the random number of rounds used by Algorithm 2 is then $R^N = \max_{m \in [M]} \max_{k \notin S_m^\star}^{N} R_{k,m}$. It is easy to prove that Lemma 4 and Lemma 10 still hold in Algorithm 2. An equivalent result to Lemma 11 can be shown

**Lemma 17.** *On event $\mathcal{E}$, for any $m \in [M]$, $k \notin S_m^\star$, $R_{k,m} \leq r_{k,m}^N$.*

*Proof.* For any $m \in [M]$, $k \notin S_m^\star$, $r = r_{k,m}$, if $k \notin B_m(r)$, then the claim is true. Otherwise, if $k \in B_m(r)$, then

$$
\begin{aligned}
\hat{\mu}'_{k,m}(r) + \Omega_{k,m}(r) \quad &\leq_{(1)} \quad \mu'_{k,m} + 2\Omega_{k,m}(r) \\
&\leq_{(2)} \quad \mu'_{k,m} + 4\widetilde{\Delta}_{k,m}(r) - 2\widetilde{\Delta}_{k,m}(r) \\
&<_{(3)} \quad \max_{i \in [K]}^N \mu'_{i,m} - 2\widetilde{\Delta}_{k,m}(r) =_{(4)} \max_{i \in B_m(r)}^N \mu'_{i,m} - 2\widetilde{\Delta}_{k,m}(r) \\
&\leq_{(1)} \quad \max_{i \in B_m(r)}^N \left( \hat{\mu}'_{i,m}(r) - \Omega_{i,m}(r) + 2\Omega_{i,m}(r) \right) - 2\widetilde{\Delta}_{k,m}(r)
\end{aligned}
$$

Then

$$
\begin{aligned}
\hat{\mu}'_{k,m}(r) + \Omega_{k,m}(r) \quad &\leq_{(2)} \quad \max_{i \in B_m(r)}^N \left( \hat{\mu}'_{i,m}(r) - \Omega_{i,m}(r) + 2\widetilde{\Delta}_{i,m}(r) \right) - 2\widetilde{\Delta}_{k,m}(r) \\
&\leq \quad \max_{i \in B_m(r)}^N \left( \hat{\mu}'_{i,m}(r) - \Omega_{i,m}(r) \right) + 2 \max_{i \in B_m(r)}^N \widetilde{\Delta}_{i,m}(r) - 2\widetilde{\Delta}_{k,m}(r) \\
&=_{(5)} \quad \max_{i \in B_m(r)}^N \left( \hat{\mu}'_{i,m}(r) - \Omega_{i,m}(r) \right) + 2 \cdot 2^{-r} - 2 \cdot 2^{-r}, \\
\implies \hat{\mu}'_{k,m}(r) + \Omega_{k,m}(r) \quad &< \quad \max_{i \in B_m(r)}^N \left( \hat{\mu}'_{i,m}(r) - \Omega_{i,m}(r) \right).
\end{aligned}
$$

where $(1)$ is using by using event $\mathcal{E}$ ; $(2)$ is using Lemma 4 ; $(3)$ uses $r = r_{k,m}^N$ and $k \notin S_m^\star$ ; $(4)$ is using event $\mathcal{E}$, and, for all $l \in S_m^\star$, $l \in B_m(r)$ ; $(5)$ holds because Lemma 10 is still valid and then, for all $j \in B_m(r)$, $\widetilde{\Delta}_{j,m}(r) = 2^{-r}$. Then $k$ is eliminated from $B_m(r)$ at round at most $r = r_{k,m}^N$, hence $R_{k,m} \leq r_{k,m}^N$. $\qquad\square$

Using this result, the sample complexity analysis is the same as for best arm identification, which yields

**Theorem 5.** *With probability $1 - \delta$, Algorithm 2 outputs the Top-$N$ arms for each agent using a total number of samples no greater than*

$$
\sup\{n \in \mathbb{N}^\star : n \leq 32 N_W^\star(\mu) \log_2(8/\Delta'_{\min})\beta^\star(n) + KM\},
$$

*where $\beta^\star(n) := \beta_\delta(n\mathbb{1}_{[M]})$.*

and we can use the same tools as in best arm identification to get an explicit upper bound depending on $N_W^\star(\mu)$.

## E   Experimental study

The general weighted collaboration bandit framework has not been studied prior to this work. We investigate its performance in the special case of federated learning with personalization [31], which corresponds to choosing weight matrices of the form $w_{n,m} = \alpha\mathbb{1}(n = m) + \frac{1-\alpha}{M}$ for any pair of agents $(n, m)$. In this special case, we propose a baseline for weighted collaborative best arm identification which is a natural counterpart of the regret algorithm proposed by [31], and compare it to our W-CPE-BAI algorithm.

### E.1   A Simple BAI Algorithm Inspired by PF-UCB

We state below as Algorithm 3 a straightforward adaptation of the PF-UCB algorithm in [30] to *personalized* federated best arm identification (BAI) ; meaning that only weight matrices of the form $w_{n,m} = \alpha\mathbb{1}(n = m) + \frac{1-\alpha}{M}$ for any pair of agents $(n, m)$ are considered. The original regret algorithm uses phased eliminations designed for each agent to identify their best arm together with *exploitation* : when all best arms have been found, or when some agent is waiting for others to finish their own exploration rounds, agents keep playing their empirical best arm. To turn this into a $\delta$-correct BAI algorithm, we remove the exploration rounds ; keep the same sampling rule within each

phase (in which the number of samples from each arm is proportional to some rate function $f(r)$) ; and calibrate the size of the confidence intervals used to perform eliminations slightly differently, introducing for any $\delta \in (0,1)$ function

$$\forall r \geq 0, B_r(\delta) := \sqrt{\frac{2 \log \left(KM\zeta(\beta)r^\beta/\delta\right)}{MF(r)}} \ .$$

where $F(r) = \sum_{p=1}^r f(p)$, for some $\beta > 1$ . In practice, we use $\beta = 2$ .

Algorithm 3, that we refer to as PF-UCB-BAI, follows the same general structure as our algorithm, with the notable difference that the number of samples of an arm $k \in B_m(r)$ in phase $r$ is fixed in advance. Under PF-UCB-BAI, when arm $k$ is still in the active set $B(r)$, agent $m$

- performs global exploration to sample it $d_{k,m}^g(r) := \lceil(1-\alpha)f(r)\rceil$ times

- and additionally performs local exploration to sample it $d_{k,m}^\ell(r) := \lceil\alpha Mf(r)\rceil$ extra times if furthermore $k \in B_m(r)$ .

Overall, $d_{k,m}(r) := d_{k,m}^g(r) + d_{k,m}^\ell(r) = \lceil(1-\alpha)f(r)\rceil\mathbb{1}(k \in B(r)) + \lceil\alpha Mf(r)\rceil\mathbb{1}(k \in B_m(r))$ new samples from arm $k$ are collected by agent $m$ during phase $r$ in order to update its estimate $\hat{\mu}_{k,m}(r)$ –the average of all available $n_{k,m}(r)$ samples for arm $k$ obtained by agent $m$– which is sent to the central server.

The mixed mean of each arm $(k,m)$ can then be computed by the server as

$$\hat{\mu}'_{k,m}(r) := \left(\alpha + \frac{1-\alpha}{M}\right)\hat{\mu}_{k,m}(r) + \frac{1-\alpha}{M}\sum_{m\neq n}\hat{\mu}_{k,n}(r) \ ,$$

and sent back to each agent. We note that, in [30], they propose that the server computes the average $\hat{\mu}_k(r) := \frac{1}{M}\sum_{m=1}^M \hat{\mu}_{k,m}(r)$ across agents, and sends this value to each agent, who can then obtain $\hat{\mu}'_{k,m}(r) := \alpha\hat{\mu}_{k,m}(r) + (1-\alpha)\hat{\mu}_k(r)$ .

Arm $k$ is eliminated from the active set $B_m(r)$ of agent $m$ if

$$\hat{\mu}'_{k,m}(r) + B_r(\delta) < \max_{j\in B_m(r)}\left(\hat{\mu}'_{j,m}(r) - B_p(\delta)\right)$$

for the confidence parameter $B_r(\delta)$. In the original algorithm, $B_r(\delta)$ is replaced by some function of $r$ and $T$, however a simple adaptation of Lemma 1 in [30] (adding a union bound on $r \in \mathbb{N}$) yields the following result. Indeed, the original result crucially exploits the sampling rule, which we did not change.

**Lemma 18.** *Event*

$$\mathcal{G} := \left\{\forall r \in \mathbb{N}^*, \forall m \in [M], \forall k \in B_m(r), \left|\hat{\mu}'_{k,m}(r) - \mu'_{k,m}\right| \leq B_r(\delta)\right\}$$

*holds with probability $1 - \delta$.*

On the good event $\mathcal{G}$ introduced in Lemma 18, observe that arm $k_m^*$ can never be eliminated from the set $B_m(r)$, therefore it has to be the guess $\hat{k}_m$ that agent $m$ outputs. This proves that Algorithm 3 is $\delta$-correct for pure exploration for the special case of federated bandit with personalization. This algorithm can therefore serve as a baseline to be compared to our proposal in this particular case.

We can also upper bound the sample complexity of this algorithm. Indeed, on event $\mathcal{G}$, like in the analysis of PF-UCB in [30], we can upper bound the number of rounds where arm $k$ is sampled by agent $m$ by $p_{k,m} := \inf\{r : B_r(\delta) \leq \Delta'_{k,m}/4\}$. When $f(p) = 2^p$, one can prove that

$$\sum_{p=1}^{p_{k,m}} f(p) = \mathcal{O}\left(\frac{\log(1/\delta)}{M(\Delta'_{k,m})^2}\right) \ .$$

Summing the (deterministic) global and local exploration cost over rounds, arms and agents, yields an exploration cost of order

$$\mathcal{O}\left(\sum_{k\in[K]}\left[\left(\frac{1-\alpha}{\min_{n\in[M]}(\Delta'_{k,m})^2}\right) + \left(\sum_{m\in[M]}\frac{\alpha}{(\Delta'_{k,m})^2}\right)\right]\log\left(\frac{1}{\delta}\right)\right) \ .$$

---

**Algorithm 3** PF-UCB-BAI

---

**Input:** $\delta \in (0,1)$, $M$ agents, $K$ arms, matrix $W$.
$f(r)$: sampling effort in phase $r$, $B_r(\delta)$: size of the confidence intervals in phase $r$.
Initialize $r \leftarrow 0$, $\forall k, m, n_{k,m}(0) \leftarrow 0$, $\forall m, B_m(0) \leftarrow [K]$, $\hat{k}_m \leftarrow 0$.
**repeat**
    # Central server
    **if** $|B_m(r)| = 1$ **then**
        $\hat{k}_m \leftarrow$ the unique arm in $B_m(r)$
        $B_m(r) = \varnothing$
    **end if**
    $B(r) \leftarrow \bigcup_{m \in [M]} B_m(r)$
    **for** $k \in B(r), m \in [M]$ **do**
        $d_{k,m}(r) = \lceil (1-\alpha)f(r) \rceil + \lceil \alpha M f(r) \rceil \mathbb{1}\,(k \in B_m(r))$
    **end for**
    Send to each agent $m$ $(d_{k,m}(r))_{k,m}$ and $d_{\max} := \max_{n \in [M]} \sum_{k \in [K]} d_{k,n}(r)$

    # Agent $m$
    Sample arm $k \in B(r)$ $d_{k,m}(r)$ times, so that $n_{k,m}(r) = n_{k,m}(r-1) + d_{k,m}(r)$
    Remain idle for $d_{\max} - \sum_{k \in [K]} d_{k,m}(r)$ rounds
    Send to the server empirical mean $\hat{\mu}_{k,m}(r) := \sum_{s \le n_{k,m}(r)} X_{k,m}(s)/n_{k,m}(r)$ for any $k \in [K]$

    # Central server
    Compute the empirical mixed means $(\hat{\mu}'_{k,m}(r))_{k,m}$ based on $(\hat{\mu}_{k,m}(r))_{k,m}$ and $W$
    *// Update set of candidate best arms for each user*
    **for** $m = 1$ **to** $M$ **do**

$$
B_m(r+1) \leftarrow \left\{ k \in B_m(r) \mid \hat{\mu}'_{k,m}(r) + B_r(\delta) \ge \max_{j \in B_m(r)} \left( \hat{\mu}'_{j,m}(r) - B_r(\delta) \right) \right\}
$$

    **end for**
    $r \leftarrow r + 1$
**until** $|B(r)| = \varnothing$
**Output:** $\{\hat{k}_m : m \in [M]\}$

---

### E.2 Numerical experiments

As we did throughout the paper, we consider Gaussian bandits with fixed variance $\sigma^2 = 1$. We denote $\hat{r}$ the average number of communication rounds across the $R$ iterations of an experiment ; $\hat{c}$ the average exploration cost of the considered algorithm across the $R$ iterations ; $\hat{\delta}$ the empirical error frequency across the $R$ iterations. $\hat{r}$ and $\hat{c}$ are reported $\pm$ their standard deviation rounded up to the closest integer, except for $\hat{\delta}$, which is rounded up to the 2nd decimal place.

We consider a synthetic instance with $K = 6$ arms, $M = 3$ agents, for $R = 100$ iterations. In order to generate randomly this instance, we sampled at random $K \times M$ values $x_{k,m}$ from the distribution $\mathcal{N}(0,1)$, and set $\mu_{k,m} = x_{k,m}/\|x\|_{\mathcal{F}}$,[9] and tested if the associated $\Delta'_{\min}$ satisfied $\Delta'_{\min} \ge 0.05$. We repeated this sampling until this condition was fulfilled.

**Comparison to PF-UCB-BAI**    Our first experiment is to compare our Algorithm 1 with the PF-UCB-BAI baseline described above. For PF-UCB-BAI we use a phase length $f(p) := 2^p \log(1/\delta)$ for $p \ge 0$. For both algorithms, we set $\delta = 0.1$ and experiment with $\alpha \in \{0.4, 0.5, 0.6, 0.7\}$. Results are reported on the left-most table in Table 1. In terms of communication cost, W-CPE-BAI (Algorithm 1) improves considerably over the baseline. Depending on the value $\alpha \in [0,1]$ (the closer it is to 1, the less agents have to communicate in order to get good estimates of their mixed expected reward) W-CPE-BAI improves or has an exploration cost which is similar to the baseline, up to a constant lower than 2.

---

[9] $\| \cdot \|_{\mathcal{F}}$ is the Frobenius norm for matrices.

Table 1: Personalized collaborative BAI with varying $\alpha \in \{0.4, 0.5, 0.6, 0.7\}$ ; the most efficient algorithm in terms of exploration cost is in bold type , all algorithms yield $\hat{\delta} = 0$ up to the $5^{\text{th}}$ decimal place (*top table*). Personalized collaborative BAI with varying $\delta \in \{0.00001, 0.0001, 0.001, 0.01, 0.05, 0.1\}$ ; ratios are rounded up to the $1^{\text{st}}$ decimal place.

| $\alpha$ | ALGORITHM | $\hat{r}$ | $\hat{c}$ | $\hat{\delta}$ |
|---|---|---|---|---|
| 0.4 | **W-CPE-BAI** | **5± 0** | **87,918± 8,634** | **0.00** |
|  | PF-UCB-BAI | 12± 0 | 109,822± 30,335 | 0.00 |
| 0.5 | **W-CPE-BAI** | **5± 0** | **75,094± 7,596** | **0.00** |
|  | PF-UCB-BAI | 11± 0 | 73,561± 17,239 | 0.00 |
| 0.6 | **W-CPE-BAI** | **5± 0** | **78,334± 7,983** | **0.00** |
|  | PF-UCB-BAI | 11± 0 | 55,812± 14,793 | 0.00 |
| 0.7 | W-CPE-BAI | 5± 0 | 76,817± 10,953 | 0.00 |
|  | **PF-UCB-BAI** | **10± 0** | **45,765± 9,591** | **0.00** |

| $\delta$ | $\hat{c}$ | $c^\star$ | $\hat{c}/c^\star$ |
|---|---|---|---|
| 0.1 | 75,094± 7,596 | 1,104 | 68.0 |
| 0.05 | 75,992± 8,468 | 1,639 | 46.4 |
| 0.01 | 80,457± 8,838 | 2,875 | 28.0 |
| 0.001 | 110,888± 31,613 | 4,643 | 23.9 |
| 0.0001 | 91,692± 9,121 | 6,411 | 14.3 |
| 0.00001 | 96,942± 9,885 | 8,180 | 11.9 |

**Comparison to an oracle algorithm**  Our second experiment is to assess the asymptotic optimality (up to some logarithmic factors) of our algorithm. In order to estimate the scaling in $T^\star(\cdot)$, we have implemented an oracle algorithm which has access to the true gaps $\left(\left(\Delta'_{k,m}\right)_{k,m}\right)$ and can compute the associated $T^\star_W(\mu)$. Then we compare the average exploration cost $\hat{c}$ with $c^\star := \left\lceil T^\star_W(\mu) \log\left(\frac{1}{2.4\delta}\right)\right\rceil$ , for $\delta \in \{0.00001, 0.0001, 0.001, 0.01, 0.05, 0.1\}$ , and $\alpha = 0.5$ . We reported the associated results in the right-most table in Table 1. We can notice that, as $\delta$ decreases, the ratio $\hat{c}/c^\star$ also decreases. As predicted by our upper bound in Theorem 2, W-CPE-BAI does not attain asymptotic optimality even for small values of $\delta$, but has a scaling to $T^\star_W(\mu) \log\left(\frac{1}{2.4\delta}\right)$ which decreases as $\delta$ goes to 0.

**Numerical considerations**  These experiments were run on a personal computer (configuration: processor Intel Core i7-8750H, 12 cores @2.20GHz, RAM 16GB). In order to solve the optimization problem defining the oracle $t(r)$, we used CVXPy [9, 1], with the commercial solver MOSEK [27] tuned to default parameters. To compute the number of samples $(d_{k,m}(r))_{k,m,r}$, we used the *optimize* module from SciPy [33]. In our experiments with this implementation of W-CPE-BAI, we found that when the instance is too hard –meaning that the associated $\Delta'_{\min}$ is small– the optimization part is subject to numerical approximation errors, which prevents the computation of the oracle allocation. This might however be mitigated by online optimization approaches, such as in [8].