# OpenReview forum: "Near-Optimal Collaborative Learning in Bandits"
_NeurIPS.cc/2022/Conference — NeurIPS 2022 Accept_

### Official Review · Reviewer_Q8Vf · 2022-07-09

**Rating:** 6
**Confidence:** 3
**Soundness:** 4 excellent
**Presentation:** 3 good
**Contribution:** 2 fair

**Summary:**

The paper proposes a collaborative bandit model in which each player seeks to identify an arm that maximizes an arbitrary weighted sum of the rewards of the players. An algorithm is proposed that is shown to be nearly optimal. The model is a generalization of a recently proposed model, for which a bounded-regret algorithm was previously given, and the paper also establishes a regret lower bound for the generalized model.


**Questions:**

Can you give a setting that is faithfully captured by the proposed model that was not already captured by a previous model?

Can you suggest a concrete family of problems beyond the ones considered in this work that might be addressed by the techniques developed here? (Note: I don't consider the top-N variant of the best arm problem in this model to be substantially more natural.)


**Limitations:**

It's fine. I'm sure that a personalized bandit algorithm could be misused to addict users to continue engaging with content platforms -- indeed, as I understand it, this is already being done. But I don't think the paper needs a "don't be evil" warning label or anything like that.


**Strengths And Weaknesses:**

The main strength of the paper is the analysis of the algorithm. The algorithm is obtained by relaxing an optimization formulation that characterizes the optimal number of rounds to identify the best arm with high probability. A similar trick is used to obtain the regret bound.

Nominally, introducing the generalized model is also a contribution. But, the main weakness of the paper is that I really cannot think of a setting that actually has the key feature of this model: namely, that the player wishes to choose an arm that does not correspond to that player's actual observed reward distribution. What would make sense to me is if the players had reward distributions that were somehow known to be similar, so that perhaps by sharing estimates they might be able to more quickly estimate their own rewards. To me, this is a more appropriate analogue of "personalization" in which the parameters (estimates) of the collective can be used as a starting point to reach a more accurate set of parameters for an individual player. (More strongly, in most collaborative bandit formulations, the problem is set up so that the players all have the same reward distributions.) But in such a case, the players can still simply solve their independent bandit instances at a possibly greater cost. The central feature in this model (like the previous "personalization" work), by contrast, is that it has been designed so that the players must communicate, since the players need to identify an arm that isn't determined by their
observed rewards. While this indeed hadn't been considered previously, in this case I think the reason is in large part the lack of motivation to consider this setting. I certainly don't agree with the claims on lines 67-78 about potential applications; in each case, the success rate for an individual user or a therapy at an individual center corresponds to the reward distribution observed for that
specific user or at that specific site. I don't see any compelling reason to choose an arm just because it has higher rewards elsewhere.

So, in summary, the question to me is whether the technical novelty outweighs the lack of a compelling motivation for the model.

---

> ### Author Response · Authors · 2022-08-01
> **Rebuttal (First part, references in the last comment)**
>
> Thank you for your review. Our research on collaborative learning in bandits was mainly motivated by improving algorithms for the federated learning with personalization setting introduced by [1] that one may or may not find realistic for real-world applications. Mathematically speaking we find it very interesting because it forces the agents to communicate to actually be able to identify their own best (mixed) arm. We then realized that our algorithms could actually be extended to a general weighted setting, which furthermore encompasses several different models studied in the growing literature on collaborative learning in bandits (see our related work section). But we believe that giving a near-optimal solution to the pure exploration problem is already interesting in the special case of federated learning in personalization. Moreover, our technical contributions could be interesting more broadly for other pure exploration problems (distributed or not), see below.
>
> >I certainly don't agree with the claims on lines 67-78 about potential applications; in each case, the success rate for an individual user or a therapy at an individual center corresponds to the reward distribution observed for that specific user or at that specific site. I don't see any compelling reason to choose an arm just because it has higher rewards elsewhere.
>
> We will elaborate in our revision (assuming an extra page) on the motivating example of an (idealized) collaborative clinical trial. We agree with you that in this context each subpopulation is aiming at finding their (local) best treatment. But solving their best arm identification in isolation may have a large sample complexity. If one is willing to assume that we have a weight matrix $W$ so that the best mixed arm $k_m^\star$ of each agent $m$ coincides with its local best arm, i.e. $\arg\max_{k} \mu_{k,m}$, then solving best arm identification in the weighted collaborative bandit could have a much smaller sample complexity due to the sharing of samples, while allowing the different clinical centers to communicate only once in a while.
>
> A first idea to build a weight matrix is to rely on a similarity function $S$ between agents and define for all $n,m$ $w_{m,n}$ to be proportional to $S(m,n)$. As an illustration, we consider the case $K=M=2$ and $\mu = [1, 0.5; 0, 0.1]$. Assume that the considered two populations are very similar ($S(1,2)=0.9$) such that the associated weight matrix W is $[0.53, 0.47; 0.47, 0.53]$. Now, solving for $K=M=2$ the oracle lower bound problem mentioned in Theorem $1$ for this weight matrix (using our implementation provided in the supplementary data: function *solve\_oracle* in *solving.py*) yields $N(\mu,W) \approx 50$. However, solving the lower bound optimization in the case where all $M$ bandit instances are independently tackled (that is, the weight matrix is the identity matrix) yields $N(\mu,Id_M) \approx 58$. Even though the actual expected values are different across the agents, the additional information about the similarity between agents allow to speed up the learning process.
>
> An alternative way to build a weight matrix W is to rely on a known clustering of the subpopulations (agents) so that the $\mu_{k,m}$ is supposed to be close to $\mu_{k,m’}$ (but not necessarily equal) for all $k$ and all agents $m,m’$ in the same cluster. Let us denote by $C_m$ the cluster to which agent $m$ belongs. Then we could set $w_{m,n} = 1(C_m = C_n)/|C_m|$ and have the mixed mean of each agent be very close to their local means. Solving the weighted collaborative BAI would lead to an improved sample complexity compared to solving individual BAI problems. Given that the means are not assumed to be exactly equal within a cluster, we note that we could not use in each cluster an algorithm for the collaborative BAI setting of [2,3] which assumes that the means are exactly equal.
>
> We agree that applying our methods for real clinical trials requires having a good matrix $W$ (e.g. a good clustering of the populations or a proper similarity measure) for which the mixed best arm coincides with a reasonable arm for each agent, which may be complex to build in practice. A possible suggestion to build this matrix in practice is to compute the similarity over a set of interesting factors, for instance, genetic markers for cancer. But of course this is only a naive suggestion, we do not claim to be experts of clinical trials.

---

> > ### Author Response · Authors · 2022-08-01
> > **Second part and references**
> >
> > >Can you give a setting that is faithfully captured by the proposed model that was not already captured by a previous model?
> >
> > Aside from the idealized example of clinical trials discussed above, in which we agree that finding a good matrix $W$ is difficult, we can mention two settings in which our general model could be useful. First, as mentioned in the paper, it allows for a slight generalization of the original federated learning with personalization framework in which each agent $m$ has a different level of personalization $\alpha_m$. Second, we could add an example of an application to marketing different subpopulations, inspired by the work of [4]. Assume a company is selling $K$ products (arms) to $M$ sub-populations (e.g., $M$ countries). Each populations is assigned a known weight $w_m$ (proportional to the importance of the corresponding market) and the company wants to find the arm $k^\star$ maximizing $\sum_{m=1}^{M} w_m \mu_{k,m}$ (to be commercialized later across all markets). It wants to do so by running in parallel a market search on the different populations, but without having the different populations communicating too often. In this particular case $w_{m,n} = w_{m}$ and $k_m^\star = k^\star$ for all $m$ (the study can therefore stop whenever one agent has found its best arm).
> >
> > >Can you suggest a concrete family of problems beyond the ones considered in this work that might be addressed by the techniques developed here?
> >
> > Classical (nearly)-optimal algorithms to solve pure exploration problems rely on either online optimization or solving a convex optimization problem after sampling an arm to estimate the optimal allocation of samples. The technique introduced in our work proposes a third way to tackle pure exploration, based on a relaxation of the initial lower bound-associated problem which leads to solving once in a while a simpler optimization problem. Our work provides a clean analysis of this approach in a concrete example but we believe it may be worth investigating for (any) other type of pure exploration problem. In particular, identifying the exact cost of this numerically appealing approach (using furthermore a small number of rounds of adaptivity) compared to exactly asymptotically optimal algorithms is an interesting theoretical question. As a concrete example of a pure exploration problem for which a relaxation of the lower bound may be available, we mention once again the work of [4]. We are also interested in investigating our lower bound relaxation approach for recent alternative models for collaborative learning exploiting similarity functions like that of [5].
> >
> > **References**
> >
> > [1] Shi, C., Shen, C., & Yang, J. (2021, March). Federated multi-armed bandits with personalization. In International Conference on Artificial Intelligence and Statistics (pp. 2917-2925). PMLR.
> >
> > [2] Tao, C., Zhang, Q., and Zhou, Y. (2019, November). Collaborative learning with limited interaction: Tight bounds for distributed exploration in multi-armed bandits. In 2019 IEEE 60th Annual Symposium on Foundations of Computer Science (FOCS) (pp. 126-146). IEEE.
> >
> > [3] Hillel, E., Karnin, Z. S., Koren, T., Lempel, R., and Somekh, O. (2013). Distributed exploration in multi-armed bandits. Advances in Neural Information Processing Systems, 26.
> >
> > [4] Russac, Y., Katsimerou, C., Bohle, D., Cappé, O., Garivier, A., and Koolen, W. M. (2021). A/B/n Testing with Control in the Presence of Subpopulations. Advances in Neural Information Processing Systems, 34, 25100-25110.
> >
> > [5] Du, Y., Chen, W., Yuroki, Y., & Huang, L. (2021). Collaborative Pure Exploration in Kernel Bandit. arXiv preprint arXiv:2110.15771.

---

### Official Review · Reviewer_cDN2 · 2022-07-10

**Rating:** 7
**Confidence:** 4
**Soundness:** 3 good
**Presentation:** 3 good
**Contribution:** 3 good

**Summary:**

Authors consider the model that there are $M$ agents and $K$ number of arms. At each round, each agent samples an arm (or is idle) and gets a reward. They assume that the agents share a known weight matrix and the objective function is to find the best arm which has a maximum expected mixed reward. The expected mixed reward of an arm for each agent is the summation of the weighted mean rewards of the arms. Therefore, the agents need to communicate with other agents to share information about their local rewards (collaborative learning).

They provide a lower regret bound in Theorem 1 for the best arm identification problem, in which the goal is to find a $\delta$-correct strategy while achieving a small exploration cost and small communication cost. Then they propose an elimination-based algorithm achieving a near-optimal exploration cost bound. The algorithm computes the number of required samples for each arm and agent, using the oracle allocation from proxy mean reward gaps. Based on the rewards, it estimated the weighted mean reward and eliminate bad arms.  With updating gap estimators, it repeats this process. Lastly, they provide a lower regret bound of this problem.


**Questions:**


- In lemma 1, I was wondering the reason why the required samples are influenced by $w^2$ rather than $w$?

- $log(1/\Delta)$ is not familiar to me in performance bounds. Could you explain why this term is hard to remove? or whether this term is significant or not for the algorithm performance?

- In practice, could we compute the oracle $P$ in definition 1 efficiently?



**Limitations:**

I believe that it would be better if the paper could provide more convincing motivation examples in which agent-dependent weighted arm rewards should be considered.

**Strengths And Weaknesses:**

**Strengths**
- They propose a novel problem in bandit research. In their problem, collaborative learning by sharing information with agents is necessary, which makes the problem non-trivial.
- They provide a lower bound for exploration cost and propose an algorithm achieving near-optimal exploration cost while requiring a small communication cost.

**Weaknesses**
- Even though the problem is novel considering collaborative learning and the paper provides some motivation examples, I have a concern about whether there exist convincing applications in the real world. This is because the purpose to find a weighted maximum arm for each agent may not be convincing in a situation where each agent can learn its own best arm individually.

- For the required samples for each agent and arm in Lemma1, it would be better if the paper provides some detailed interpretation of the influence of weight value on the number of samples such as where $w^2$ comes from.

- Also, it would be better to discuss how to compute oracle $P$ (definition1) efficiently to be used in the algorithm.

- In the regret bound in Theorem 2, $log(1/\Delta)$ could be large in the worst case, which makes the bound far from the lower bound.

---

> ### Author Response · Authors · 2022-08-01
> **Rebuttal**
>
> Thank you for your review. We provide below the answer to your three questions:
>
> - The factor $w^2$ comes from the proof of the lower bound for pure exploration (in Appendix) when solving the associated constrained optimization problem through the KKT conditions. Further intuitions into the scaling of sample complexities with $w^2$ rather than $w$ can be gained from the upper bound perspective. Note that the confidence interval width $\Omega_{k,m}(r)$ scales with $\sqrt{\sum_{m=1}^M\frac{w^2_{n,m}}{n_{k,n}(r)}}$. This becomes more clear in the case of normal random variables, where the confidence interval width scales with the standard deviation. The quantity $\sqrt{\sum_{m=1}^M\frac{w^2_{n,m}}{n_{k,n}(r)}}$ is the standard deviation of $X_{k,m}$ provided $n_{k,n}$ observations from random variables $X_{k,n}$, assuming the $(X_{k,n})_{k,n}$ are normal.
>
> - The $\log(1/\Delta_{\min})$ multiplicative factor is a consequence of the phased structure of the algorithm, and a similar term is also present in the analysis of other phased algorithms, see, e.g. [1,2]. The final number of arm pulls for each (k,m) pair is its maximum over rounds. Using a phased algorithm in our setting, the sample complexity seems to inevitably scale with the number of rounds.
>
> - Computing the (relaxed) oracle requires to solve K optimization problems
> $$\min_{\tau^k \in (\mathbb{R}^+)^{M}} \sum_{(k,m) \in [K] \times [M]} \tau^k_{m} \textnormal{ s.t. } \forall m, \sum_{n \in [M]}\frac{w_{n,m}^2}{\tau^k_{n}} \leq (\Delta'_{k,m})^2/2$$
> at fixed k. These optimization problems belong to the family of disciplined optimization problems which can be solved using the CVXOPT Python package for instance (as done in our implementation). We will add in the main text the details currently provided in the numerical considerations discussed in Appendix E.2.
>
> Regarding the practical relevance of the proposed model which is mentioned as a weakness, please read our answer to reviewer Q8Vf.
>
> **References**
>
> [1] Fiez, Tanner, et al. "Sequential experimental design for transductive linear bandits." Advances in neural information processing systems 32 (2019). See Theorem 2. The $\log(1/\Delta_\min)$ term can be tracked in the derivations at the bottom of page 15.
>
> [2] Du, Yihan, et al. "Collaborative Pure Exploration in Kernel Bandit." arXiv preprint arXiv:2110.15771 (2021). See Theorem 1. The $\log(1/\Delta_\min)$ term can be tracked in the derivations at the bottom of page 23.

---

> > ### Comment · Reviewer_cDN2 · 2022-08-05
> > **Response to authors**
> >
> > Thanks for your detailed response to my comments.
> > All my concerns are resolved.

---

### Official Review · Reviewer_MREF · 2022-07-11

**Rating:** 7
**Confidence:** 4
**Soundness:** 4 excellent
**Presentation:** 3 good
**Contribution:** 4 excellent

**Summary:**

This paper introduces a collaborative learning multi-armed bandits formulation, which generalizes the previous stochastic federated bandit with personalization framework. The authors provide two new lower bounds on the sample complexity of pure exploration and on the regret. They also propose a near-optimal algorithm for pure exploration, achieving the lower bound up to logarithmic factors.

**Questions:**

No major questions.

**Ethics Review Area:**

["I don’t know"]

**Limitations:**

The authors did a great job explaining the limitation in the theory side, and the work does not have negative societal impact.

**Strengths And Weaknesses:**

Strengths:

- The proposed model is more flexible in terms of how the rewards are mixed, compared with the previous work (Shi et al, 2021a).

- Novel lower bound analyses are provided, which largely answer the open question proposed in Shi et al (2021a). The analysis technique is inspiring not only for federated bandits but also for similar cooperative multi-agent bandits models.

- Inspired by the lower bound analysis, a new algorithm termed W-CPE-BAI is proposed.

- For the best arm identification formulation, the proposed algorithm is shown to be capable of approaching the lower bound.

- For the regret minimization formulation, a regret lower bound is proved, which largely complements the conjecture in Shi et al (2020a) and further sheds light on the corresponding algorithm design.

Weaknesses:
- Nothing major.

---

> ### Author Response · Authors · 2022-08-01
> **Thank you for the review**
>
> Thank you for your positive review.

---

### Meta-Review · Area_Chair_qHaN · 2022-09-03

**Recommendation:** Accept
**Confidence:** Certain

**Metareview:**

The paper introduces a new formulation for a multi-player MAB motivated by federated learning. The reviews are all positive and agree that the paper is a significant contribution.

**Award:**

No

---

### Decision · Program_Chairs · 2022-09-14

Accept